# Long-read assembly of the Chinese rhesus macaque genome and identification of ape-specific structural variants

Yaoxi He[1,2,3,4,8], Xin Luo[1,2,3,4,8], Bin Zhou[1,2,3,4,8], Ting Hu[1,2,3,4,8], Xiaoyu Meng [1,2,3,4,8], Peter A. Audano[5], Zev N. Kronenberg[5], Evan E. Eichler [5,6], Jie Jin [7], Yongbo Guo[1,2,3,4], Yanan Yang[1,2], Xuebin Qi [1,2,3] & Bing Su[1,2,3]

We present a high-quality de novo genome assembly (rheMacS) of the Chinese rhesus macaque (*Macaca mulatta*) using long-read sequencing and multiplatform scaffolding approaches. Compared to the current Indian rhesus macaque reference genome (rheMac8), rheMacS increases sequence contiguity 75-fold, closing 21,940 of the remaining assembly gaps (60.8 Mbp). We improve gene annotation by generating more than two million full-length transcripts from ten different tissues by long-read RNA sequencing. We sequence resolve 53,916 structural variants (96% novel) and identify 17,000 ape-specific structural variants (ASSVs) based on comparison to ape genomes. Many ASSVs map within ChIP-seq predicted enhancer regions where apes and macaque show diverged enhancer activity and gene expression. We further characterize a subset that may contribute to ape- or great-ape-specific phenotypic traits, including taillessness, brain volume expansion, improved manual dexterity, and large body size. The rheMacS genome assembly serves as an ideal reference for future biomedical and evolutionary studies.

[1] State Key Laboratory of Genetic Resources and Evolution, Kunming Institute of Zoology, Chinese Academy of Sciences, Kunming 650223, China. [2] Primate Research Center, Kunming Institute of Zoology, Chinese Academy of Sciences, Kunming 650223, China. [3] Center for Excellence in Animal Evolution and Genetics, Chinese Academy of Sciences, Kunming 650223, China. [4] Kunming College of Life Science, University of Chinese Academy of Sciences, Beijing 100101, China. [5] Department of Genome Sciences, University of Washington School of Medicine, Seattle, WA 98195, USA. [6] Howard Hughes Medical Institute, University of Washington, Seattle, WA 98195, USA. [7] Nextomics Biosciences, Wuhan 430000, China. [8] These authors contributed equally: Yaoxi He, Xin Luo, Bin Zhou, Ting Hu, and Xiaoyu Meng. Correspondence and requests for materials should be addressed to B.S. (email: sub@mail.kiz.ac.cn)

Rhesus macaque (*Macaca mulatta*) is the most widely studied nonhuman primate (NHP) in biomedical science[1], and the closest human relative where gene editing has been approved for generating animal models. This makes it an indispensable species for understanding human disease and evolution, and as a result, a high-quality rhesus macaque reference genome is a prerequisite. The captive-born Chinese and Indian rhesus macaques represent two subspecies populations with moderate genetic differentiation (diverged ~ 162,000 years ago[2]). Previous efforts using next-generation sequencing (NGS) have established a draft genome assembly of the Indian rhesus macaque[1,3], but the current genome version (rheMac8) is incomplete with > 47,000 gaps (~72 mega-base pairs (Mbp) in length). Similarly, only a draft genome of the Chinese rhesus macaque was available[4] (contig N50: 13 Kbp; total gap length: 331 Mbp). Structural variants (SVs) are known to be important in primate evolution and disease[5,6]. However, owing to the poor contiguity (fragmentation) and incompleteness (many gaps) of the current macaque genome assemblies, it has been difficult to systematically identify SVs since detection methods based on assemblies cannot identify SVs in regions where the assembly fails.

Long-read sequencing and multiplatform scaffoldings provide an opportunity for assembling high-quality genomes—in part owing to long reads (> 10 Kbp), more-uniform sequence error distribution of long-read sequencing (such as PacBio), and the availability of multiple scaffolding technologies (such as Bionano and 10X Genomics)[7–9]. Compared with the short-read data from NGS, the long-read data are useful in resolving complex genomic regions, such as the highly repetitive and GC-rich regions. Long-read sequencing data, in particular, increase sensitivity for the identification and sequence resolution of SVs[7,8].

The emergence of apes during evolution (Hominoidea: a branch of Old World tailless anthropoid primate native to Africa and Southeast Asia) required a series of evolutionary innovations, including taillessness[10], large body size[11], increased brain volume/complexity (especially in great apes and humans)[12,13], and improved manual dexterity[14]. Apes are the sister group of Old World monkeys, together forming the catarrhine clade. Recently, using PacBio data, we reported the high-quality assembly of great-ape genomes, including potentially adaptive human-specific SVs[7]. A high-quality genome of an Old World monkey species (e.g., rhesus macaque), however, was lacking and is required as an outgroup to identify and characterize the functional genetic changes that occurred in the common ancestor of the ape lineage.

Here, we present the first high-quality Chinese rhesus macaque genome (rheMacS) de novo assembly using long-read sequencing and multiple scaffolding strategies and contrast it with the Indian macaque genome (rheMac8). We identify 53,916 SVs in rheMacS, and by comparative genomic approaches, we focus on 17,000 ape-specific structural variants (ASSVs), identifying potentially functional SVs that may contribute to the major phenotypic changes during ape evolution. The rheMacS assembly provides one of the most complete Old World monkey reference genomes to date and an important genetic resource to the biomedical community.

## Results

**De novo assembly of the Chinese rhesus macaque genome**. We chose an adult male Chinese rhesus macaque (*Macaca mulatta*) and extracted high-molecular-weight DNA from peripheral blood. We first sequenced and assembled the genome using SMRT long-read sequencing (100-fold genome coverage; average subread length of 9.7 Kbp) and FALCON (Supplementary Fig. 1, Supplementary Table 1, 2 and Methods). We then scaffolded the contigs utilizing Bionano data (101-fold genome coverage) and generated de novo optical genome map. We filled the remaining gaps in the assembled genome using PBJelly[15] and corrected the errors of the PacBio long reads using Arrow[16]. To further improve the accuracy of the genome assembly, Illumina whole-genome shotgun (WGS) short-read data (50-fold genome coverage) and Pilon[17] were used to correct remaining errors in the sequence contigs (Supplementary Table 2 and Supplementary Fig. 2). Finally, the Hi-C data (105-fold genome coverage) was used to anchor scaffolds to chromosome models (Supplementary Table 3 and Methods).

For the purpose of gene annotation, we extracted total RNA from 16 tissues, including large intestine, lung, epididymis, liver, testis, muscle, bladder, prefrontal cortex (PFC), cerebellum, skin, spleen, kidney, stomach, small intestine, heart, and pancreas (Methods). We performed full-length transcriptome sequencing (also called isoform sequencing, Iso-Seq) by pooling the RNA samples from the ten tissues (Supplementary Table 1) and generating 100 Gbp of Iso-Seq data (Methods). To assess expression of the novel gene models, we also generated short-read RNA sequencing (RNA-seq) for each of the 16 tissues, producing 185 Gbp data (9.04~13.92 Gbp for each tissue).

In summary, we have generated a de novo assembly of the Chinese rhesus macaque genome (rheMacS) that represents 2.95 Gbp of the chromosomes with contig N50 length of 8.19 Mbp and scaffolds N50 length of 13.64 Mbp (Table 1). Compared with the previous rhesus assembly (rheMac8), the rheMacS shows much less fragmentation (348,493 vs. 4741 sequence contigs, > 98% reduction in total contig numbers), improving sequence contiguity by 75-fold (contig N50) (Table 1, Fig. 1, and Fig. 2a) and scaffold N50 length by threefold (Table 1 and Supplementary Fig. 3).

**Quality assessment**. We used the assembled rheMacS genome to first close gaps in the rheMac8 reference genome (Methods). We filled 21,940 of remaining N-gaps in rheMac8 adding 60.81 Mbp of additional sequences (2% of the entire genome, Table 1, Supplementary Fig. 4 and Supplementary Table 4). As expected, 75% of the closed gaps consisted of various classes of repeat DNA: long interspersed nuclear elements (LINEs), short interspersed nuclear elements (SINEs), and simple short-tandem repeats (Supplementary Fig. 5 and Supplementary Table 6). Notably,

**Table 1 Comparison of assembly statistics between rheMac8 and rheMacS**

| Assembly | rheMac8 | rheMacS |
|---|---|---|
| Assembly approach | WGS and BAC | WGS and Hi-C |
| Sequencing platform | Sanger, Illumina | PacBio, Bionano, Hi-C, Illumina |
| Number of contigs | 348,493 | 4741 |
| Contig N50 (Mbp) | 0.11 | 8.19 |
| Number of scaffolds | 286,263 | 4543 |
| Scaffold N50 (Mbp) | 4.19 | 13.64 |
| Number of gaps[a] | 47,882 | 2858 |
| Total gap length (Mbp)[a] | 72.05 | 5.46 |
| Total bases (bp)[b] | 2,835,963,390 | 2,955,490,605 |
| Ungapped bases (bp)[c] | 2,763,913,633 | 2,950,026,318 |

The assembly statistics of rheMac8 were obtained from NCBI (accession number: GCA_000772875.3). *WGS* whole-genome shotgun, *BAC* bacterial artificial chromosome
[a]Only N-base regions in the assembled chromosomes were counted
[b]Only chromosome-placed bases were counted
[c]Non-N bases in the assembly

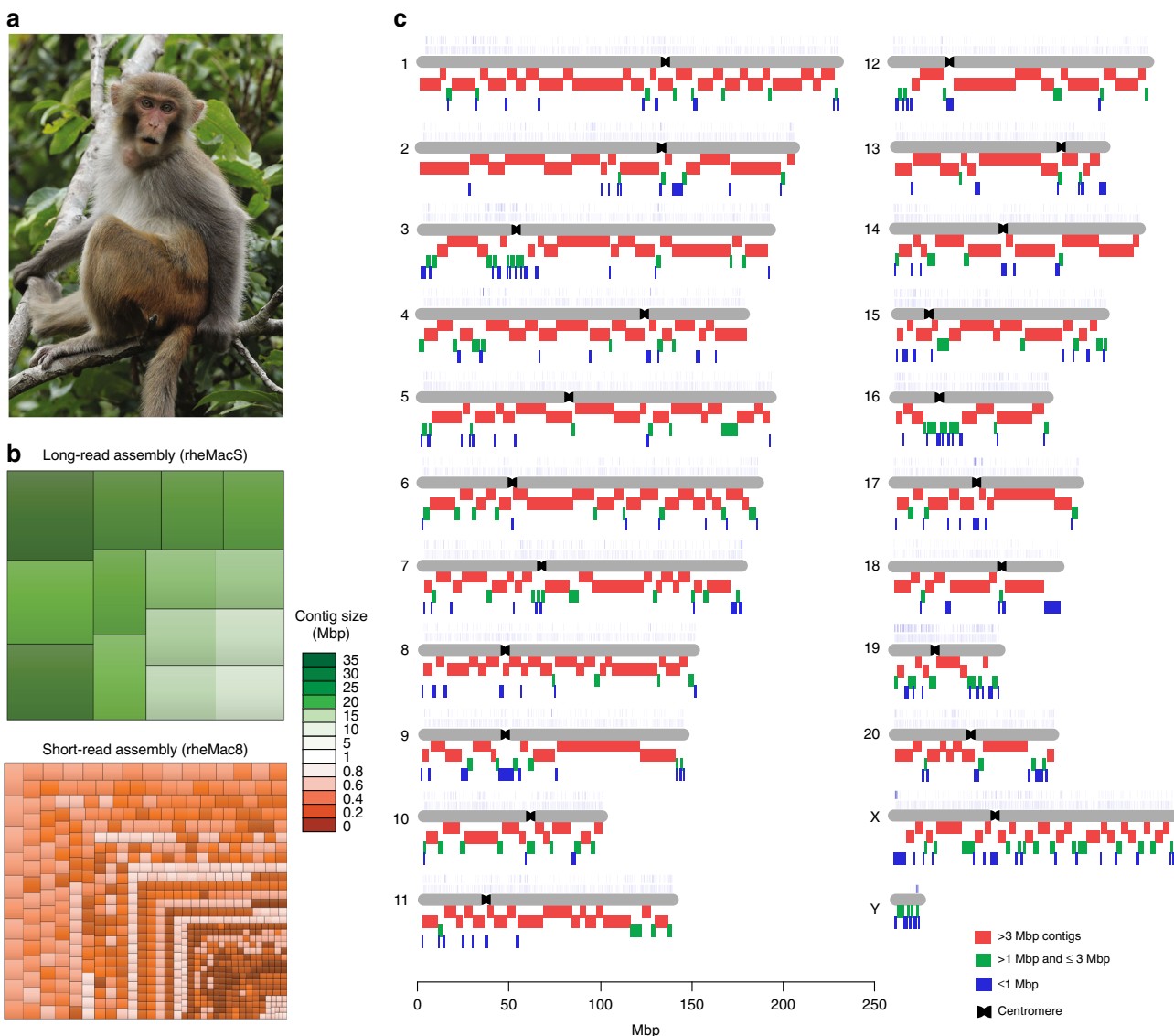

**Fig. 1** Long-read assembly of the Chinese rhesus macaque genome. **a** Chinese rhesus macaque (*Macaca mulatta*) (photograph courtesy by Jiaxin Zhao). **b** Treemaps for fragmentation difference between long-read and short-read rhesus assemblies. The rectangles represent the largest contigs that account for ~ 300 Mbp (~ 10%) of the assembly. **c** The chromosomal distribution of contigs of the rheMacS genome assembly. We compare genome sequence contiguity between rheMac8 and rheMacS. Thousands of gaps in rheMac8 were closed by longer contigs from rheMacS. The assembled contigs include > 3 Mbp (red), between 1 Mbp and 3 Mbp (green), and those < 1 Mbp (blue). The small contigs (< 1 Mbp) tend to consist of either centromeric or telomeric sequences. The centromeres of each chromosome are indicated based on previous annotation[30]

7146 of the filled gaps (13.27 Mbp) map within genes and 41 map to the coding regions adding 10.17 Kbp of protein-encoding sequences (Fig. 2b and Supplementary Table 4, 5).

The rheMacS assembly has far fewer gaps (rheMacS: 2858 vs. rheMac8: 47,882), which are shorter in length (rheMacS: 5.46 Mbp vs. rheMac8: 72.05 Mbp) when compared with rheMac8 (Fig. 2c and Table 1). To assess the consensus accuracy of rheMacS, we aligned each chromosome sequence of rheMacS to rheMac8. More than 98% of the assembled sequences are concordant by length and orientation (Supplementary Fig. 6 and Supplementary Table 7). To evaluate sequence accuracy, we also mapped the 50-fold Illumina short-read data to the rheMacS assembly as the previous studies described[18]. The estimated quality value (QV) score was 50 ($1.11 \times 10^{-5}$ variants per base) (Supplementary Table 8 and Methods). This is well below one error per 10,000 bases, a quality standard used for human genomes[19].

Next, we compared the mappability between rheMacS and rheMac8 by deep sequencing genomes from five additional unrelated Chinese rhesus monkeys (Illumina HiSeq X10 PE-150bp, 50 × depth per individual). We mapped the short-read data to rheMacS and rheMac8, respectively, observing a higher mapping rate for rheMacS compared with rheMac8 (99.60% vs. 99.24%, respectively; $P < 0.001$, two-tailed paired $t$ test) (Fig. 2d and Supplementary Table 9). Genome-wide variant calling using the WGS data and different tools (GATK and SAMtools) for the five monkeys shows a similar number of SNV types (SNPs and INDELs) and SV types (deletion, insertion, duplication, and inversion, SV length ≥ 50 bp) (Supplementary Table 10). These results suggest that rheMacS will be useful for future variant detection.

Interestingly, when we mapped the RNA-seq short reads from the 16 tissues (~ 37 million reads per tissue) to rheMacS and rheMac8, we observed a significantly increased mapping rate for

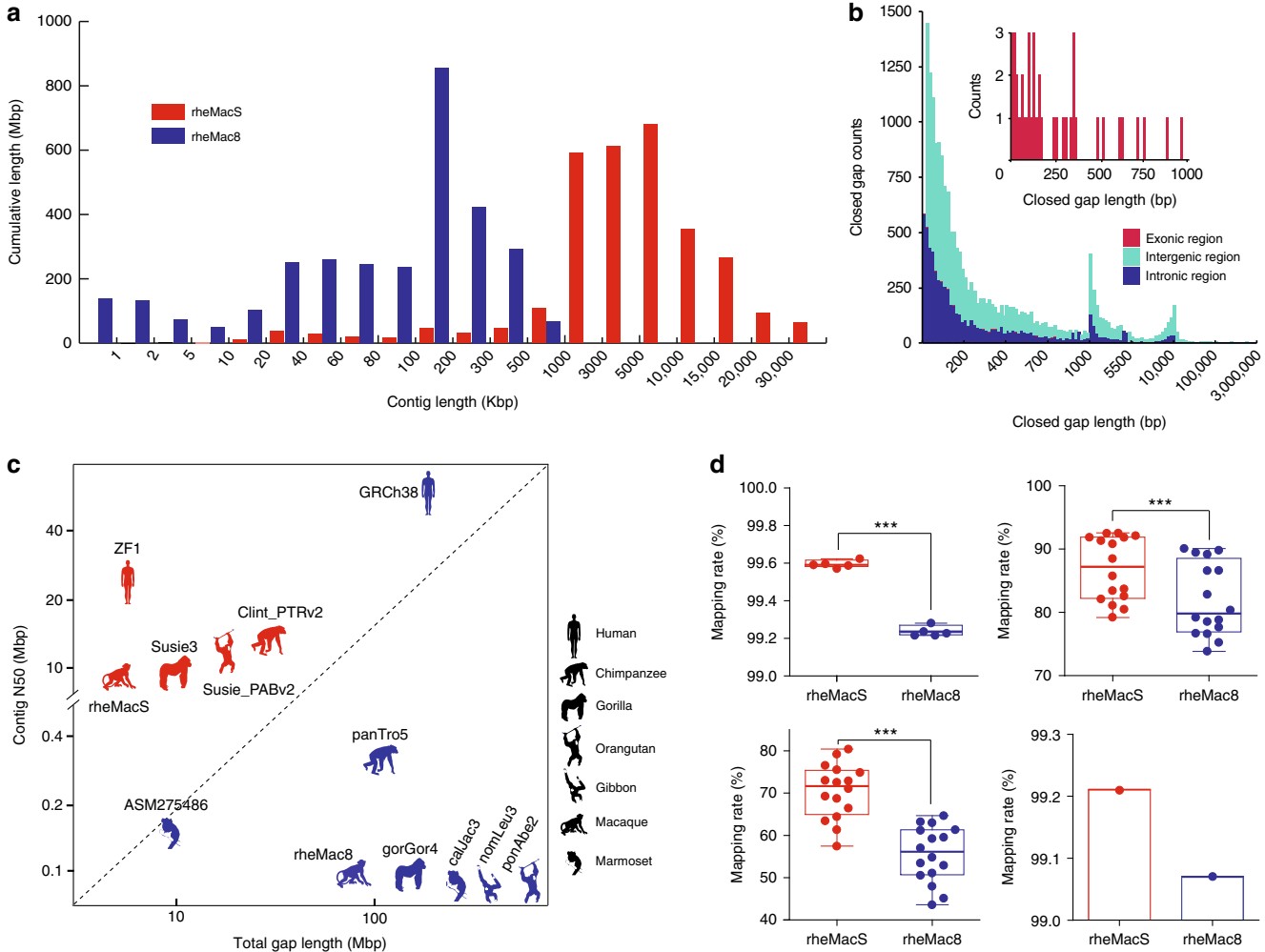

**Fig. 2** Quality assessment of the Chinese rhesus macaque genome assembly. **a** Comparison of sequence contig length distribution between rheMacS and rheMac8. **b** Length distribution of the closed gaps in rheMac8. **c** Comparison of assembly quality among various reported primate genome assemblies. Genomes assembled with long PacBio long reads (red) are compared against those assembled using Illumina short-read and Sanger sequencing data (blue). **d** Comparison of mapping rates when short-read NGS data (upper left), RNA-seq data (by Hisat2: upper right; by Bowtie2: lower left), and Iso-Seq data (lower right) are mapped to rheMacS and rheMac8, respectively. Two-tailed paired *t* test was used for statistical assessment. ***$P < 0.001$. The boxplot shows the mean value (central line), upper and lower quartiles (bounds of box) and min/max values (whiskers)

rheMacS compared with rheMac8 (86.87% vs. 81.96%, respectively; $P < 0.001$, two-tailed paired *t* test) (Fig. 2d and Supplementary Data 1). Long-read Iso-Seq data show similar mapping rates (99.21% for rheMacS vs. 99.07% for rheMac8) (Fig. 2d and Supplementary Table 11). Although some of these differences may relate to divergence between Chinese and Indian origin of the genomes, our results suggest that the rheMacS assembly will also enhance future rhesus monkey transcriptomic analyses.

**Gene annotation**. We performed ab initio gene annotation for rheMacS using the 100 Gbp of long-read (Iso-Seq) sequence data (2,468,473 full-length non-chimeric (FLNC) transcripts with mean length of 2759 bp) (Supplementary Fig. 7), and the 185 Gbp of short-read RNA-seq data generated from the 16 tissues (Methods). We annotated in total 20,389 protein-coding genes in rheMacS, just slightly fewer than rheMac8 (20,605) (Supplementary Table 12). The analysis, however, generated more iso-forms in rheMacS (288,773 in rheMacS vs. 276,000 in rheMac8) (Supplementary Table 11) where the average length was longer for rheMacS (1564 bp in rheMacS vs. 1489 in rheMac8) (Supplementary Fig. 8 and Supplementary Table 12). Improved gene annotation results, in part, from rescued missing exons and better

coverage of full-length transcripts after gap closures (Supplementary Table 5). Comparison of orthologous gene families between rheMacS and other primate genome assemblies (rheMac8, apes, and mouse) indicated similar gene family composition (Supplementary Fig. 9), suggesting a reliable annotation quality for rheMacS. We also annotated repeats by RepeatMasker[20] and Tandem Repeats Finder[21] (Methods). We found that 1.5 Gbp (54.04%) of the rheMacS genome were annotated as repeats, similar with the repeat ratios in rheMac8 and ape genomes (Supplementary Table 13 and Supplementary Data 2).

To further assess gene annotation, we benchmarked 4104 universal single-copy orthologs (BUSCO)[22], which by design should be uniformly present in all mammalian genomes. We find that 3836 BUSCO genes (93.5%) are completely annotated in rheMacS. There are 191 BUSCO genes (4.7%) with fragmented annotation and only 77 BUSCO genes (1.8%) not annotated in rheMacS, contrasting the higher missing rate in rheMac8 (3.1%) (Supplementary Table 14). We also annotate noncoding RNA (ncRNA) genes in rheMacS (Methods), obtaining 49,698 long noncoding RNA (lncRNA), 11,035 microRNA (miRNA), 544 ribosomal RNA (rRNA), 2373 small-nucleolar RNA (snRNA), and 718 transfer RNA (tRNA) (Supplementary Table 15).

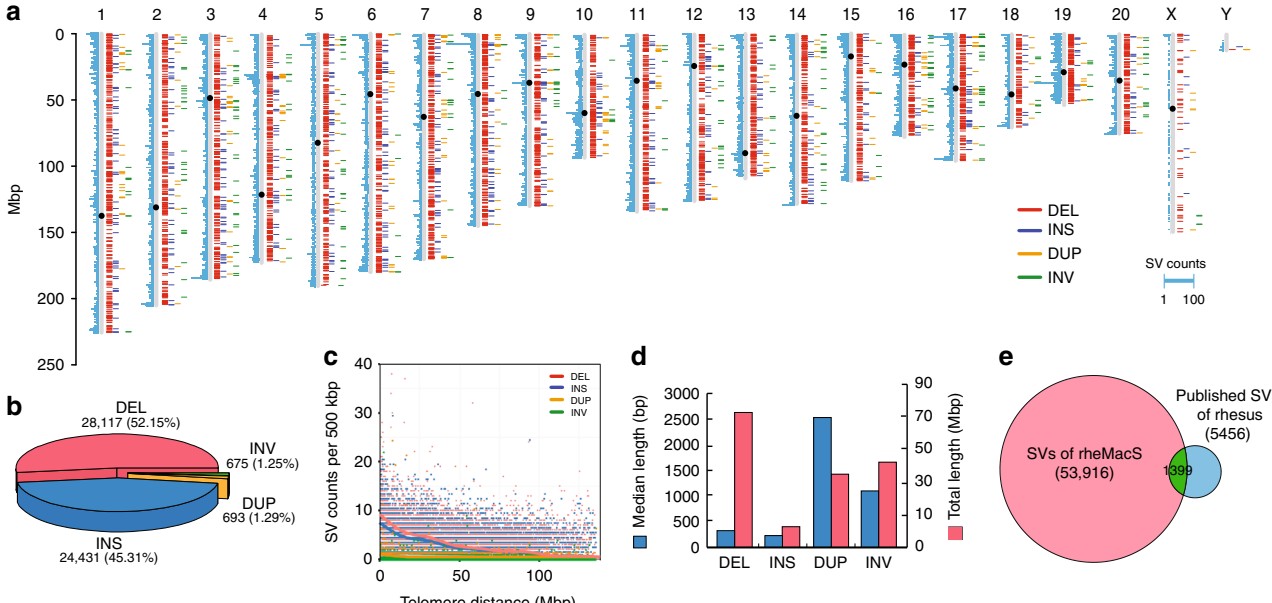

**Fig. 3** Structural variants (SVs) in rheMacS. **a** The distribution of large SVs (≥ 1 Kbp) among the rhesus macaque chromosomes. The histogram marks on each chromosome (light blue on the left) indicate the counts of SVs based on per 500 Kbp windows. The black dot on each chromosome indicates the centromere position. **b** The percentages of the four SV types including deletions (DEL), insertions (INS), duplications (DUP), and inversions (INV). **c** The SV distribution along with the increase of telomere distance. The SVs are counted with a sliding window size of 500 Kbp. The multicolor dots refer to the four-type SV counts in a 500 Kbp bin, and the solid lines indicate the distribution of average counts. **d** The length statistics of the rheMacS SVs. **e** Overlaps of the rheMacS SVs with previously reported SVs by aCGH and NGS data. The overlap cutoff is set to require > 50% reciprocal overlapping of SV length

**Identification of SVs in the rheMacS genome assembly**. Given the increased sensitivity for SV detection and their increased probability of affecting gene expression and phenotype[7], we identified putative SVs using a read-mapping approach[23] by mapping long reads against the Indian macaque genome assembly (rheMac8). We defined SVs as variants ≥ 50 bp in size and identified 53,916 SVs in the rheMacS assembly (Fig. 3). The set includes 28,117 deletions, 24,431 insertions, 693 duplications, and 675 inversions (Supplementary Data 3, Methods). These rheMacS SVs correspond to 26% (1399/5456) of the previously reported SVs from Indian rhesus macaques using array genomic hybridization and Illumina short-read genome sequencing[24–26]. Notably, we estimated that 96% (51,919/53,916) of the SVs are novel, highlighting the advantage of long-read sequencing data for genome-wide SV detection (Fig. 3e and Supplementary Data 3).

The genomic distribution of the identified SVs in rheMacS is not random, and they tend to cluster in the proximal telomere regions (Fig. 3a and Fig. 3c), consistent with the pattern seen in our recent report of long-read human genomes[27]. We estimate a 1.8-fold increase in SV density within the subtelomeric regions (5 Mbp region of each chromosome end, $P < 10^{-16}$, permutation test). The median lengths for deletion, insertion, duplication, and inversion are 319 bp, 231 bp, 2575 bp, and 1119 bp, respectively. Combined, the SVs span 170.8 Mbp and affect 5.78% of the entire genome (Fig. 3d). Using VEP (Variant Effect Predictor), we find that 2% (1069 SVs) map within exons, 39% (21,009 SVs) in introns, and the remaining 59% (31,838 SVs) in intergenic regions. The newly identified >50,000 SVs by long-read sequencing data will be a useful resource in studying primate genome evolution.

We genotyped these 53,916 SVs in five unrelated Chinese rhesus macaques with Illumina WGS data (Illumina HiSeq X10 PE-150bp, 50 × depth per individual) using SVTyper[28] (Methods). The results showed that 42,126 (78.13%) SVs were present in these macaques, among which 5654 (13.42%) were fixed and 36,472 (86.58%) were polymorphic (Supplementary Figure 10 and

Supplementary Data 3). There were 11,790 (21.87%) invalidated SVs, and the majority of them (59.58%) were located in repeat regions, such as LINEs, SINEs, and simple repeats, suggesting that short-read data are limited to resolve SVs in repetitive regions. In particular, 94.67% of the invalidated SVs were insertions, highlighting the limitation of NGS short-read data in detecting novel insertions[29].

**Detection of ASSVs**. Sequencing of both ape and macaque using the same long-read sequencing platform provides an opportunity to identify, for the first time, ASSVs that emerged in the ape lineage since divergence from the Old World monkeys. Using smartie-sv[7,9], we performed genome-pairwise comparisons among the rheMacS assembly and three published great-ape long-read genome assemblies (chimpanzee, orangutan, and gorilla)[7] as well as a human long-read assembly (ZF1) (Supplementary Figure 11 and Supplementary Table 16, Methods). To exclude SVs that occurred in the rhesus macaque lineage, we used the published genome assembly of common marmoset (assembly ID: ASM275486), which is the latest and most complete genome assembly of a New World monkey species (contig N50: 155 Kbp; total gap length: 9.5 Mbp) (Fig. 2c). Considering the poor quality of the current gibbon genome assembly (contig N50: 35 Kbp; total gap length: 205 Mbp) (Fig. 2c), we only used it when conducting local sequence alignment for manual check of the candidate ASSVs in order to distinguish ASSVs from great-ape-specific SVs (GASSVs).

Using the above strategy, we detected 17,000 candidate ASSVs after the filtering (Methods), including 13,456 deletions and 3544 insertions. ASSVs account for only 7.97 Mbp (on average 0.26% of the ape genomes) (Fig. 4 and Supplementary Data 4). As expected, the majority of the ASSVs (78.05%) map to common repeats in the genome such as SINEs, LINEs, and LTRs (Supplementary Table 17). We find that 30 ASSVs consist of tandem repeats and likely are the result of non-allelic homologous recombination although further work will be required to delineate

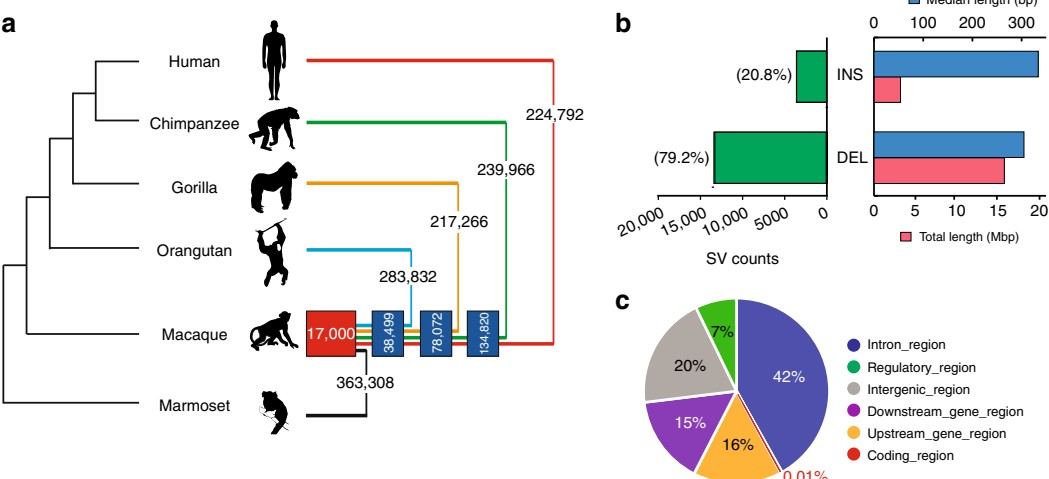

**Fig. 4** Summary of the ape-specific structural variants (ASSVs). **a** The cladogram depicts the phylogenetic relationship among the studied primate species, including human, three great apes, rhesus macaque, and common marmoset. The numbers of the identified SVs by genome-pairwise comparisons are indicated. The numbers in the blue boxes represent the overlap for different pairwise results. The number in the red box indicates the identified 17,000 ASSVs. Gibbon (lesser ape) was not included due to the poor quality of the published genome assembly. We used the common marmoset genome assembly generated by short-read sequencing data to filter the SVs that occurred in the rhesus monkey lineage (bottom panel). **b** Statistics of ASSVs. **c** Pie plot for ASSV annotation

mutational mechanisms (Supplementary Data 4). The ASSV chromosomal distribution is concordant with the known sequence synteny relationships between rhesus macaque and apes (Supplementary Fig. 12 and Supplementary Table 18)[30].

We first annotated the identified ASSVs using VEP according to the human GRCh38 coordinates (Methods). In total, 12,255 ASSVs map either within or near (5 Kbp flanking transcriptional start and end) 3412 coding genes with the remaining 4745 ASSVs mapping to intergenic regions (Fig. 4c). We explored functional enrichment among the 3412 ASSV-related genes using DAVID GO (gene ortholog) analysis and pathway analysis. Although no functional category reached statistical significance, we observe several interesting functional categories among the top 5% (10/208) categories (Supplementary Fig. 13 and Supplementary Data 5). Cilium assembly and morphogenesis ranked the top one category in the GO biological process and FAC (functional annotation clustering, enrichment score = 3.13)—categories that play a critical role for development of the vertebrate nervous system and in regulating neuronal cell fate, migration, and differentiation[31,32]. We, therefore, speculate that ASSVs in this functional category might contribute to great-ape brain-related phenotypic changes (e.g., expansion in brain size).

**ASSVs in gene-coding regions**. There are 25 ASSVs (17 deletions and 8 insertions) located in the coding sequences of 32 genes (Supplementary Data 6). We tested these 25 ASSVs by PCR and Sanger sequencing, and 16 of them were validated as true ASSVs and 9 were false ASSVs (Supplementary Data 6). Among the 16 validated coding ASSVs (9 deletions and 7 insertions) (Table 2), 6 were annotated as NMD (nonsense-mediated decay) variants[33], 4 as splice site variants, 2 as in-frame deletions, 2 as frameshift variants, and 2 as coding variants. These SVs affect 15 genes, which are involved in brain function/neuro-diseases (*IL20RB*, *NMNAT3*, *CLCN3*, *FBF1*, *ZNF563*, *PPP1R15A*, and *PTK6*)[34–38], bone development (*VPS33A*, *CLCN3*, and *DECR2*)[39,40], spermatogenesis/sexual hormone (*EXOSC10* and *RUVBL2*)[41,42], immunodeficiency (*UNG*)[43], fatty-acid degradation (*DECR2*)[44] and skin disease (*SLC17A9*)[45] (Table 2). For example, there is an in-frame deletion ASSV (318 bp deletion) in *CCDC168*, a gene with unknown function (Supplementary Fig. 14 and Supplementary

Data 6). There are three ASSVs disrupting the splice acceptor/donor sites, resulting in lineage-specific protein (3076 bp insertion in *NMNAT3*) (Supplementary Fig. 15 and Supplementary Data 6) and lineage-specific transcript isoforms (316 bp insertion in *EXOSC10* and 1435 bp insertion in *IL20RB*) (Supplementary Fig. 16 and Supplementary Data 6). Functionally, *IL20RB* and *NMNAT3* are involved in neuroprotection[34,36]. *EXOSC10* is associated with the regulation of male germ cells[41].

**ASSVs in regulatory elements functioning in the brain**. As the majority of the identified ASSVs map to intronic or intergenic regions, their potential functional impact may relate to gene expression regulation. Utilizing previously published brain ChIP-seq data from human, chimpanzee, and rhesus macaque[46], we searched for ASSVs mapping to enhancers where corresponding genes showed ape-specific distances, i.e., human and chimpanzee have similar enhancer activities while rhesus monkey shows significantly lower or higher activities ($P < 0.05$, two-tailed unpaired $t$ test). We first identified 7155 ape-monkey differential enhancers (ADEs) in eight brain regions (Supplementary Data 7, Methods), where we observe significant differences of H3K27Ac signal (marker of enhancer activity) between apes (human and chimpanzee) and rhesus macaques in at least one brain region. When overlapping the 17,000 ASSVs with these ADEs, we identify 87 ADEs corresponding to 111 ASSVs, which have the potential to affect the regulation of 65 nearby genes (Supplementary Fig. 17 and Supplementary Data 8).

Among the 111 ASSVs, 21 (14 deletions and 7 insertions) are high confident based on manual curation (Supplementary Data 8), which affect 20 ADEs (Fig. 5a). Interestingly, except for two ADE showing monkey-ape difference in more than one brain region, the other 18 ADEs show interspecies differences in only one brain region. With respect to enhancer activity, 13 ADEs are enhancer gains, where apes show significantly stronger enhancer signals than macaque, and the other five ADEs are enhancer losses (Fig. 5a and Supplementary Data 8). Notably, the ape-gained enhancers are dominant in four of the five brain regions except for PcGm (precentral gyrus), which contained three ADEs—all of which resulted in enhancer loss in the ape lineage. We observed the same pattern when including the 111 candidate ASSVs

**Table 2 The 16 validated ASSVs located in gene-coding regions**

| Chrom[a] | Start[a] | End[a] | Length | Type | Gene | Exon | Consequence | Gene functions[b] |
|---|---|---|---|---|---|---|---|---|
| chr1 | 11095424 | 11095738 | 316 | INS | EXOSC10 | 4/5 | Splice_acceptor_variant | Spermatogenesis |
| chr3 | 136986715 | 136988148 | 1435 | INS | IL20RB | 3/7 | Splice_donor_variant | Neuroprotective and intelligence |
| chr3 | 139580804 | 139583878 | 3076 | INS | NMNAT3 | 5/8 | Splice_acceptor_variant | Neuroprotective; intelligence, and axonal protection |
| chr4 | 169663104 | 169663246 | 144 | INS | CLCN3 | 2/14 | Coding_sequence_variant | Schizophrenia, autism spectrum disorder; bone pattern, and synaptic transmission |
| chr12 | 109105086 | 109105279 | 195 | INS | UNG | 6/7 | NMD_transcript_variant | Immunodeficiency; epididymitis, and somatic hypermutation |
| chr12 | 122264731 | 122264829 | 100 | INS | VPS33A | 2/14 | NMD_transcript_variant | Lysosome function; foot abnormality and acetabular dysplasia |
| chr13 | 102742592 | 102742592 | 318 | DEL | CCDC168 | 4/4 | Inframe_deletion | <Unknown> |
| chr16 | 405554 | 405554 | 66 | DEL | DECR2 | 3/10 | NMD_transcript_variant | Fatty-acid degradation; hemoglobin concentration and body mass index |
| chr17 | 75909754 | 75909754 | 304 | DEL | FBF1 | 29/29 | NMD_transcript_variant | White matter hyperintensity; brain and eye measurements and cilium assembly |
| chr17 | 75909803 | 75909803 | 631 | DEL | FBF1 | 29/29 | NMD_transcript_variant | White matter hyperintensity; brain and eye measurements and cilium assembly |
| chr19 | 12318606 | 12318606 | 317 | DEL | ZNF563 | 4/4 | Frame_shift | Intelligence; schizophrenia and educational attainment |
| chr19 | 48874925 | 48874925 | 285 | DEL | PPP1R15A | 1/1 | Inframe_deletion | White matter disease and Alzheimer's disease |
| chr19 | 49004220 | 49004220 | 337 | DEL | RUVBL2 | 4/15 | NMD_transcript_variant | Chorionic gonadotropin level; follicle stimulating hormone level; cell cycle and mitotic |
| chr20 | 62965134 | 62965134 | 564 | DEL | SLC17A9 | 9/13 | Splice_region_variant | Porokeratosis; cutaneous photosensitivity; papule and pruritus |
| chr20 | 63533689 | 63533689 | 537 | DEL | PTK6 | 4/8 | Frame_shift | Dysgraphia; schizophrenia, autism spectrum disorder and hair shape |
| chr22 | 39964324 | 39964903 | 578 | INS | Z82206.1 | 2/2 | Coding_sequence_variant | <Unknown> |

Chrom chromosome, INS insertion, DEL deletion, NMD nonsense-mediated decay
[a]The coordinates are based on human GRCh38
[b]The gene functions are collected from GeneCard database and literatures[35,37–40,43–45]

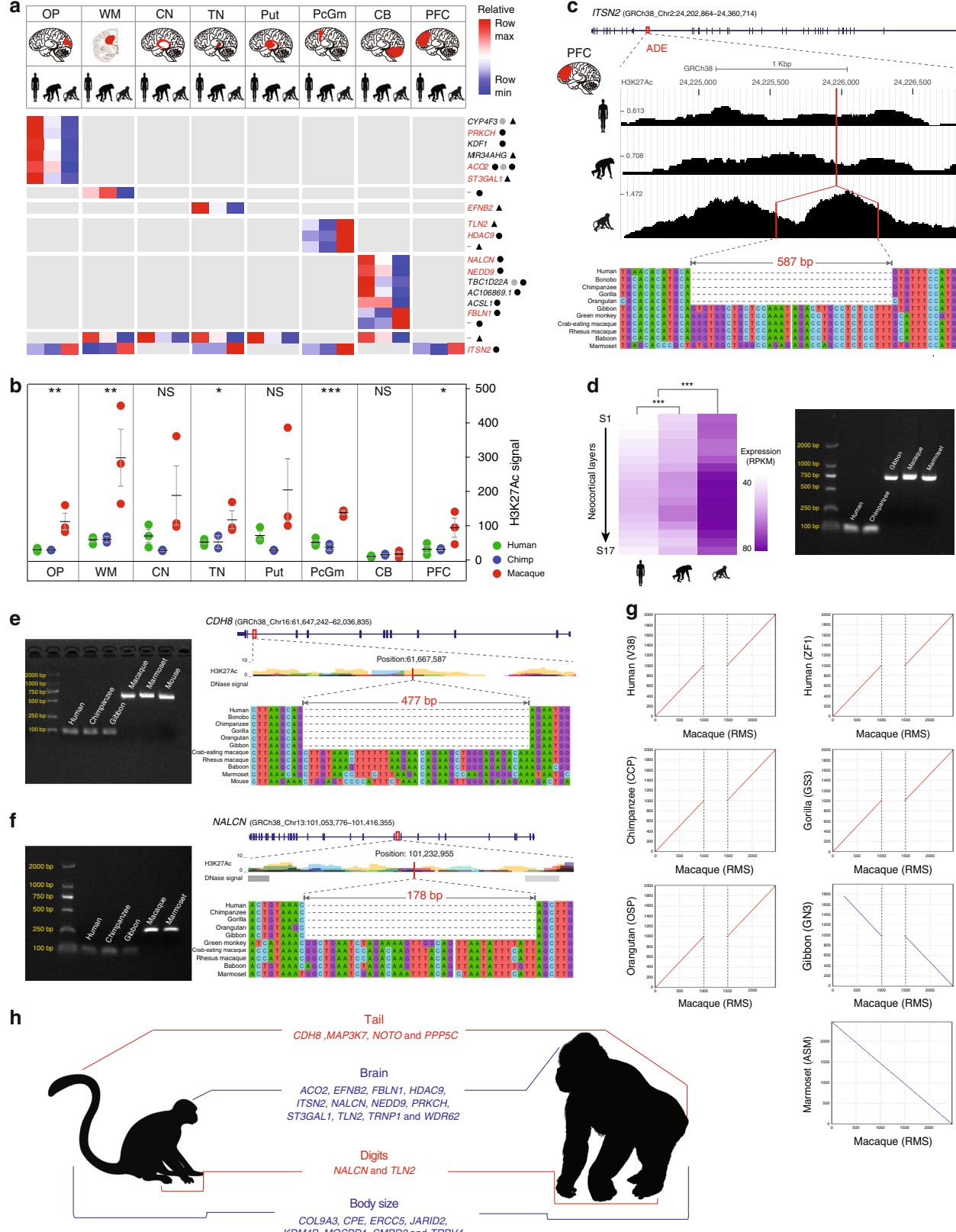

(Supplementary Fig. 17). Interestingly, we identify 10 ADEs (containing 10 ASSVs) near 10 genes, which are related with neuronal cell function, brain function, and neurological diseases (Fig. 5a and Supplementary Data 8).

We experimentally validated two ASSVs using PCR and Sanger sequencing. The 587 bp deletion that disrupts an ADE (GRCh38: chr2:24,224,548-24,226,881) in five brain regions of great apes in *ITSN2* (Fig. 5b, Supplementary Fig. 18, and Supplementary

**Fig. 5** ASSVs associate with ape-specific (ASPs) or great-ape-specific phenotypes (GASPs). **a** Heatmap illustrating the ADEs with high-confident ASSVs in eight brain regions. The nearest genes are indicated and the corresponding brain regions are indicated in red. The neurofunction-related genes are highlighted in red. ASSV deletions (circles) and insertions (triangles) are denoted. The circles/triangles in black and gray refer to high-confident ASSVs and candidate ASSVs, respectively. **b** Comparison of H3K27Ac signals of the ADE with an ASSV (587 bp deletion) in *ITSN2* among human, chimpanzee, and macaque. The ADE exhibits significant signal difference between human/chimpanzee and macaque in five brain regions (*$P < 0.05$; **$P < 0.01$; ***$P < 0.001$; NS-not significant, $P > 0.05$). **c** A 587 bp deletion within intron-29 of *ITSN2* disrupts a putative enhancer sequence in the great-ape lineage, with reduced enhancer activity in human and chimpanzee compared with rhesus macaque. The H3K27Ac signals in PFC and sequence alignments are shown. **d** *ITSN2* exhibits significantly lower expression in chimpanzee and human compared with macaque with 16 neocortical layers (1S–16S) and the adjacent white matter (17 S) (left panel). PCR validation is shown (right panel). **e** A 477 bp deletion located in intron-10 of *CDH8*, a gene related to tail development. **f** A 178 bp ape lineage-specific deletion in intron-12 of *NALCN*, a gene associated with human fetal adducted thumbs. **g** A dot plot alignment highlights a 587 bp deletion (located in *ITSN2*) in apes compared with rhesus macaque and marmoset. **h** Summary of candidate ASSVs/GASSVs located in genes associated with ASPs (in red) or GASPs (in blue). A statistical assessment of the H3K27Ac signals and gene expression difference was conducted using two-tailed unpaired and paired $t$ test, respectively. NS: not significant ($P > 0.05$) and ***$P < 0.001$

Data 8), which encodes Intersectin-2, affecting clathrin-mediated endocytosis and is critical in synaptic vesicle recycling in neurons[47]. In the PFC, according to the published transcriptome data of 16 PFC layers and the adjacent white matter[48], the expression of *ITSN2* is significantly lower in humans and chimpanzees than in rhesus macaques, consistent with their reduced H3K27Ac signals (an indication of enhancer activity) owing to the ASSV that disrupts an ADE in *ITSN2* (Fig. 5c, d, g). Another 1128 bp deletion is a cerebellum CB-specific ADE (GRCh38: chr6:11,262,523-11,265,271) of apes in *NEDD9* (neural precursor cell expressed, developmentally downregulated 9), a gene relevant for dendritic spine maintenance, cognitive ability[49], and hindbrain development[50] (Supplementary Fig. 19 and Supplementary Data 8). This ASSV leads to increased enhancer activities of *NEDD9* in cerebellum of humans and chimpanzees (Supplementary Figure 19). Gene expression data of cerebellum in NHPs are not available to check its effect on *NEDD9* expression.

**ASSVs associated with ape-specific or great-ape-specific phenotypic (GASP) traits.** There are a set of ape-specific phenotypes (ASPs) that are not found among other primates, including tail-lessness[10] and improved manual dexterity[14]. In addition, brain capacity expansion[12,13] and large body size are GASPs[11,51]. To understand whether the identified ASSVs might contribute to these ASPs or GASPs, based on their known functions from databases and published literature, we compiled a list of 451 genes related to these phenotypes (137 genes for tail development, 19 genes for brain size regulation, 56 genes for adducted thumbs and 239 genes for body size; Supplementary Table 19 and Methods).

We performed a simulation to test the randomization of ASSVs for these ASPs or GASPs (see Methods for details). The results showed significance for body size ($P = 6.0 \times 10^{-6}$) and brain size ($P = 3.81 \times 10^{-4}$) but not manual dexterity ($P = 0.88$) or tail development ($P = 0.90$) (Fisher's exact test). This result suggests that there is an enrichment of ASSVs associated with GASPs.

To search for candidate ASSVs that potentially affect the 451 ASP/GASP-related genes, we conducted a detailed survey in each gene set of the four traits. Among the 137 tail development-related genes (Supplementary Table 19, Methods), we identified 32 ASSVs and 4 of them are high-confident ASSVs (Supplementary Data 9). For example, we found a 477 bp deletion located in *CDH8* (Fig. 5e). The mouse *Cdh8* knockout showed abnormal tail movements[52]. Another example is a 130 bp deletion in *MAP3K7* (Supplementary Fig. 20). The *MAP3K7* knockout mouse had a shorter tail. Importantly, the knockout mouse also showed an abnormal form of vertebral bodies (HP:0003312) and abnormal dental morphology (HP:0006482)[53], both of which show diverged phenotypes between monkeys and apes[51]. In addition, we observed H3K27Ac peaks (human data) in these two ASSV

regions (from ENCODE databases; Fig. 5e and Supplementary Fig. 20), implying that these ASSVs may contain regulatory elements such as enhancers. Notably, both genes belong to the Wnt-signaling pathway, a crucial pathway for tail extension and axial termination in vertebrates[54]. Interestingly, the two genes are also highly expressed in the nervous system (brain and spinal cord) and are involved in body plan and somitogenesis during embryo development.

The increase in ape manual dexterity involves a series of anatomical changes of the hand, including the evolution of flat nails and more-sensitive finger pads, especially those associated with opposable thumbs[14]. Among the 56-known adducted-thumb associated genes (ATA genes) (Human Phenotype Ontology ID: HP:0001181), we identify 12 ASSVs (6 deletions and 6 insertions) in 3 ATA genes of which 4 are high confident (Supplementary Table 19 and Supplementary Data 9). For example, an ASSV (a 178 bp deletion) that maps to the intron-12 of *NALCN* (sodium leak channel, non-selective) exhibited enhancer signals. It was reported that *NALCN* was associated with human fetal adducted thumbs[55] (Fig. 5f).

It is known that great apes have a larger body size compared with Old World monkeys and lesser apes[11,51]. Among the 239 genes associated with body size (Supplementary Table 19), we identified three genes (*CPE*, *COL9A3*, and *ERCC5*) with GASSVs (Supplementary Figs. 21, 22). For example, *COL9A3* and *ERCC5* associate with a series of disorders showing growth failure and short stature[56,57], and embryonic and postnatal growth retardation was previously reported in gene knockout mice[58]. These genes may affect body size in the great-ape lineage as these GASSVs are shared among all great apes (Supplementary Figs. 21, 22, Supplementary Table 19 and Supplementary Data 9).

Taken together, we identify a set of ASSVs/GASSVs as candidate genetic changes associated with ape-specific or great-ape-specific traits that can serve as a resource to study the genetic basis of phenotypic innovations that emerged during ape/great-ape evolution (Fig. 5h). However, owing to the lack of ENCODE data of the correspondent tissues in NHPs, the speculated regulatory roles of these ASSVs in the ape lineage are yet to be validated.

## Discussion
Rhesus macaque serves as an indispensable species for understanding human biology. Here, using long-read sequencing and multiple scaffolding technologies, we generated a high-quality genome assembly of rhesus macaque. The rheMacS genome assembly greatly improves the contiguity and completeness of the current version of the rhesus macaque reference genome (rheMac8). Using the rheMacS assembly, we characterized a large number of SVs, most of which were not observed in previous studies using array and NGS platforms. In particular, through

comparative genomic analyses, we discovered 17,000 candidate ASSVs that may contribute to the emergence of ape-specific or great-ape-specific traits (such as taillessness, brain size, adducted thumbs, and body size) during evolution (Fig. 5h).

Among the 17,000 identified ASSVs, ~ 80% are deletions, and only 20% are insertions, indicating that losses of genomic sequences are more frequent than sequence gains, consistent with the results from the NGS data of great apes[59]. Notably, 78.05% of the ASSVs contain repeat elements such as SINEs, LINEs, and LTRs (Supplementary Table 17), suggesting that repeat-element-mediated mutations are likely the main source of SVs. It should be noted that we only focused on the functional inference of two common SV types (deletions and insertions) because they account for the great majority (> 98%) of SVs in the genome assemblies. Also, the ASSV filtering using NGS-based genome data of gibbon and marmoset can introduce unambiguity when calling the other SVs types such as duplications and translocations. These SVs types (duplications, inversions, translocations, and chromosomal rearrangements) may also contribute to phenotypic evolution in primates, and they are worth studying in the future.

We focused on understanding the functional implications of ASSVs, disrupting gene-coding integrity and those located in regulatory elements, which may contribute to ape-specific or great-ape-specific traits. We validated 16 ASSVs located in the coding regions of 15 genes, and 7 of them are associated brain functions and neurologic diseases (*IL20RB*, *NMNAT3*, *CLCN3*, *FBF1*, *ZNF563*, *PPP1R15A*, and *PTK6*)[34–38]. For the ASSVs in the noncoding regions, we evaluated their potential functional effects based on the published brain ChIP-seq and transcriptome data. For example, we identified a 587 bp great-ape-specific deletion in an ADE of *ITSN2*, which may explain the attenuated enhancer activity in human/chimpanzee compared with macaque in five brain regions (OP, WM, TN, PcGm, and PFC) (Fig. 5b). Consistently, human and chimpanzee exhibited lower expression than macaque in the PFC and the adjacent white matter (Fig. 5d, Supplementary Fig. 18). Previous studies showed that over-expression of *ITSN2* resulted in the inhibition of transferrin uptake and the blockage of clathrin-mediated endocytosis, which is critical in synaptic vesicle recycling in neurons[47] and dendritic spine development[60]. Interestingly, *ITSN2* knockout mice displayed decreased grip strength and vertical activity (MGI:5631233), a trait that showed difference between apes and monkeys[51].

Primate evolution is an intricate process, and the changes of anatomical features from monkeys to apes may interact with each other during ontogeny and scale properly with body size as the organism grows. For example, most of the skeletal traits correlate strongly and positively with body size[51]. In particular, the tailless trait of apes strongly linked with body size, positional behavior and bone structure, etc. Although it is hard to tell the order of trait emergence, it is possible that the related traits may be controlled by linked genetic changes during evolution. For example, the 178 bp deletion in *NALCN* (a gene related with adductive thumbs) disrupted an ADE in the cerebellum and exhibited a sharp difference in enhancer activities between monkeys and apes (Fig. 5f and Supplementary Data 8). It was reported that *NALCN* was also associated with neurodevelopmental diseases and the mutation of *NALCN* leads to syndromic neurodevelopmental impairment[61]. Another example is the ASSV (242 bp deletion) within an ADE in PcGm in *TLN2* (Fig. 5a), a brain function-related gene[62]. This gene was also reported as a cause of fifth finger Camptodactyly[63]. At present, the ENCODE data in NHPs are mostly from the brain, whereas the data from other tissues are not available. Given regulatory elements such as enhancers usually function in a tissue-specific manner, there is a great need

to generate ENCODE data of NHPs so that the potential functions of the identified ASSVs associated with the key ape-specific traits other than the brain (such as taillessness and body size) can be annotated.

We used the marmoset assembly to filter out the monkey-lineage-specific SVs. Because of the poor assembly quality of marmoset, the discrepancy in assembly quality among the reference genomes of marmoset and rhesus (rheMacS) make it difficult to filter SVs unambiguously. We found that 54.51% (9266/17,000) ASSVs are uncertain (Supplementary Data 4 and Methods). We conducted a manual check of 311 ASSVs, among which 16.4% (51 ASSVs) were high-confident ASSVs, and 18.3% (57 ASSVs) were false ASSVs. The remaining 65.3% (203 ASSVs) ASSVs were uncertain because the assembly and the sequence quality around the genomic regions of these ASSVs were poor in marmoset (Supplementary Data 7, 8, 9). Similarly, when using the gibbon assembly to distinguish the GASSVs from ASSVs, we encountered the same situation. In contrast, we experimentally tested the 25 coding ASSVs, and 16 (64%) of them were validated by PCR (Supplementary Data 6). Also, except for the two non-coding ASSVs that cannot be reliably confirmed by the Sanger sequences due to their completely repetitive nature (100% sequence identity), we experimentally validated 20 noncoding ASSVs located in predicted regulatory elements (e.g., enhancers) of genes associated with ape-specific or great-ape-specific phenotypic traits. The higher validation rate of these ASSVs is likely owing to the relatively conserved and well-resolved sequences in the genome assemblies. Nevertheless, the high-quality (preferably long-read) genome assemblies of New World monkeys (e.g., marmoset) and gibbons are needed to capture all lineage-specific SVs in apes.

High-quality genome assemblies are a prerequisite for comparisons among species. Most of the current primate genome assemblies were based on NGS short reads or shotgun Sanger sequencing with limited resolution for SV detection. For example, we observed a significantly poor reciprocal validation when comparing the NGS genome with the long-read assembled genome of rhesus macaque (Supplementary Fig. 23). Consequently, we should upgrade the current primate genome assemblies using long-read sequencing and multiple scaffolding techniques, a critical step forward in order to heighten the utility of NHP models for biomedical research and to greatly promote the understanding of primate evolution.

## Methods
**Sample information**. An adult Chinese rhesus macaque (*Macaca mulatta*) (male, 5-year-old) was used for tissue sample collection in this study.

**Data generation**. For long-read sequencing data, we extracted the high-quality genomic DNA from fresh blood samples of the macaque. The libraries with an average insert size of 20 Kbp were constructed using SMRTbell Template Prep Kits and then sequenced on a PacBio Sequel instrument at the Genome Center of NextOmics Bioscience Co., Ltd (Wuhan, China). A total of 53 SMRT cells were run on the PacBio Sequel system and 299.6 Gbp (subreads) of data (with 100-fold coverage of genome) for rheMacS were generated. The average length and the N50 length of long subreads are 9.7 Kbp and 14.7 Kbp, respectively.

For Bionano optical mapping, the Bionano Genomics (BNG) instrument Saphyr was used to generate optical molecules using restriction enzyme Nt.BspQI. High-molecular-weight DNA (> 200 Kbp) was extracted from whole blood, and we constructed a high-quality sequencing library according to the recommended protocols. A total of 304 Gbp of clean data were generated with, on average, 8.6 labels per 100 Kbp, and the molecular reads N50 length was 216.4 Kbp (Supplementary Fig. 24).

For Hi-C data, two Hi-C libraries were prepared[64]. In brief, five million cells were fixed with formaldehyde and lysed, and the cross-linked DNA was digested with a restriction endonuclease. The sticky ends of the digested fragments were biotinylated and proximity ligated to form chimeric junctions. Biotinylated DNA fragments were enriched and sheared to a size of 200–300 bp for preparing the paired-end sequencing libraries. Libraries were sequenced on an Illumina HiSeq

X10, In total, we generated 308 Gbp (PE-150bp) data with 103-fold genomic coverage.

For long-read RNA sequencing (Iso-Seq), total RNA was extracted from 10 tissues (heart, liver, spleen, lung, kidney, muscle, brain, epencephala, testicle and stomach) using a TRIzol extraction reagent (ThermoFisher), subjected to the Iso-Seq protocol with different ratios of three library sizes (1–6 Kbp, 1–4 Kbp, and 4–6 Kbp), and sequenced by the PacBio Sequel sequencer.

For short-read DNA sequencing, DNA samples were subjected to TruSeq DNA library kit (Illumina) and sequenced on the Illumina Sequencer HiSeq X10 with paired-end 150 bp sequence reads. In total, 162 Gbp of data were generated.

For short-read RNA sequencing, RNA samples were subject to the TruSeq mRNA library kit (Illumina) and sequenced on the Illumina HiSeq X10 sequencer. In total, 185 Gbp data were produced in total. Total RNA was extracted from 16 tissues (large intestine, lung, epididymis, liver, testis, muscle, bladder, PFC, skin, spleen, kidney, stomach, small intestine, cerebellum, heart and pancreas) using a TRIzol extraction reagent (ThermoFisher).

**De novo genome assembly**. We estimated genome size by k-mer distribution analysis with the program Jellyfish (v1.1.11)[65] ($k = 17$) by the Illumina short reads. The 17-mer curve exhibits a significant heterozygosity peak and repeat peak for rheMacS (Supplementary Fig. 25), and the estimated genome size of rheMacS was ~ 3.07 Gbp, and the heterozygosity of the genome was ~ 0.5% (Supplementary Fig. 25).

FALCON (v1.8.7) (https://github.com/PacificBiosciences/FALCON) was used for de novo assembly with PacBio long reads by the following steps: (i) raw subreads overlapping for error correction; (ii) preassembly and error correction; (iii) overlapping detection of the error-corrected reads; (iv) overlap filtering; (v) constructing a graph from overlaps; and (vi) constructing contigs from graph. After error correction, where a length cutoff of 14 Kbp was used for initial seed reads mapping, and then the error-corrected reads were used to construct the assembly graph and to generate a draft assembly with a genome size of 3 Gbp with a N50 length of 4.75 Mbp.

Hybrid genome assembly was performed by the Bionano Saphyr software with the manufacturer-recommended parameters. We constructed an optical map by running the hybrid scaffold pipeline (Bionano Solve v3.1). Subsequently, we filled the gaps based on the scaffold level genome. All PacBio long reads were aligned to the scaffold genome by PBJelly software[15].

To further improve the accuracy of the genome assembly, a two-step polishing strategy was used for the initial assembly. The initial polishing was performed with Arrow[16] using PacBio long reads only. Because of the high error rate of PacBio raw reads, we also used Pilon (v1.20)[17] to further improve the PacBio-corrected assembly with the highly accurate Illumina short reads.

Scaffolds within each chromosomal linkage group were then assigned using the Hi-C-based proximity-guided assembly. The original cross-linked long-distance physical interactions were then processed into paired-end sequencing libraries (Supplementary Fig. 26). First, all the reads from the Hi-C libraries were filtered by the HiC-Pro software (v2.8.1)[66], and then the paired-end reads were uniquely mapped onto the draft assembly scaffolds, which were grouped into 22 chromosome clusters and scaffolded using LACHESIS software[67] (Supplementary Table 20).

**Gap closure**. We closed the gaps in the reference genome of rhesus monkey (rheMac8) using the approach of a previous study. A region consisting of continuous runs of Ns ($N > 1$) in the rheMac8 chromosomes was defined as a gap. We extracted 47,882 rheMac8 gaps (Supplementary Table 4) based on the BED file format, and the 5 Kbp of flanking sequences upstream and downstream of the gaps were mapped to the assembly by MUMmer[68] (nucmer -f -r -l 15 -c 25). We defined a gap as closed when the following two criteria were met: (i) the two flanking sequences of a gap in rheMac8 could be both aligned to the rheMacS assembly and the aligned length of the flanking sequences is > 2.5 Kbp and (ii) when recording the rightmost coordinates (RC) and leftmost coordinates (LC) of rheMacS, non-N bases were observed between LC and RC in rheMacS. The total bases between LC and RC were counted as the closed-gap length.

We evaluated the repeat content of filled gaps, which were mapped to whole-genome repeats (annotated by rheMacS). An effective overlap was defined with overlapping length reaching at least 50% of the reciprocal similarities (Supplementary Table 6). In addition, we annotated the filled gaps by mapping to the annotation of rheMacS, including exonic, intronic, and intergenic regions (Supplementary Tables 4, 5).

**Assembly quality assessment**. Consensus quality: to evaluate the concordance between rheMacS and rheMac8, MUMmer was used to perform pairwise alignment and calculate the sequence identity (nucmer --mum -c 1000 -l 100).

Sequence quality: we mapped all 50 × depth Illumina short reads to the rheMacS assembly using the BWA-MEM module[69]. Then, we used Picard to mask the PCR duplicates and generated the dedup.bam file. Variants were called by the Genome Analysis Toolkit (GATK, v3.6) HaplotypeCaller module. The SNPs and INDELs were filtered using the GATK VariantFiltration module with the following criteria, respectively: SNPs filtering: "QUAL < 50; QD < 2.0; FS > 60.0; MQ < 30.0;

MQRankSum < − 12.5; ReadPosRankSum <−8.0; DP < 15"; INDELs filtering: "QUAL < 50; QD < 2.0; FS > 200.0; ReadPosRankSum < − 20.0; DP < 15". As previous studies described[18], we used QV score to assess the sequence quality, which represents a per-base estimate of accuracy and is calculated as $QV = −10 \log 10(P)$ where $P$ is the probability of error. We counted the total number of the homozygous SNVs (SNPs + INDELs), which represent the sites with base error in the rheMacS assembly. The base-error rate was calculated as the number of homozygous sites divided by the total size of the rheMacS assembly (Supplementary Table 8).

Mappability: 50 × depth WGS data were generated by Illumina sequencer HiSeq X10 (PE-150 bp) for five rhesus monkeys, which was used to evaluate the mappability and availability for variant calling. We used FastQC and GATK to perform quality control (QC) for the Illumina WGS reads, and then we mapped these short reads to rheMacS and rheMac8 assemblies, respectively, using the BWA-MEM module. SAMtools and GATK standard pipeline were used to call variants. Delly (v0.7.7)[70] was used to call SVs. All underlying algorithm, tool versions, and command lines for variant calling and filtering are listed in Supplementary Table 21.

**Gene annotation**. For gene annotation, 119.01 million RNA-seq reads and 1,275,860 Iso-Seq reads were generated with a mean length of 2791 bp (Supplementary Fig. 27).

Gene models were constructed with MAKER (v2.31.8)[71], which incorporated the ab initio prediction, the homology-based prediction, and the RNA-seq assisted gene prediction. For the ab initio gene prediction, the repeat regions of the rheMacS genome assembly were first masked based on the result of repeat annotation, and then SNAP[72] and Augustus (v3.2.2)[73] were trained for model parameters from homolog genes of BUSCOs, and they were then employed to generate gene structures. BUSCO (v3.0.1) was used to conduct the mammalian BUSCO analysis[22] with the Mammalia odb9 set of 4,104 genes. BUSCO was run against all complete protein-coding sequence sets of annotation in rheMacS.

For ncRNA annotation, the published RNA database Rfam (http://rfam.xfam.org/) was used to predict rRNA, snRNA, and miRNA by sequence alignments. The tRNAscan-SE and RNAmmer tools were used to annotate tRNA and rRNA, respectively. LncRNAs were predicted using ncrna_pipeline (https://bitbucket.org/arrigonialberto/lncrnas-pipeline) (Supplementary Fig. 28).

**Transcript analysis**. After obtaining the raw Iso-Seq data from the PacBio Sequel system, we performed data QC and identified the full-length reads (by cDNA primer detection and removal: existence and completeness of 5′ primer, 3′ prime and poly-A). After this procedure, we obtained 2,468,473 FLNC reads. The size distribution of the FLNC reads, which correspond to candidate full-length transcripts, are displayed in Supplementary Fig. 7. To further improve the accuracy of isoform sequence, we made clustering for the FLNC reads using ICE (iterative clustering algorithm) and obtained the unpolished consensus reads. Then both these unpolished consensus reads and non-full-length reads were finally polished using Arrow[16]. Based on the Arrow consensus predicted accuracy, only sequences with accuracy of > 99% were deemed high quality. The statistics of consensus reads are shown in Supplementary Table 22. To further improve the base accuracy of consensus reads, we made error correction with short-read RNA-seq data using LoRDEC[74]. The number of NGS-corrected reads is listed in Supplementary Table 22.

To compare the transcripts of rheMacS and rheMac8 all Iso-Seq FLNC reads and ICE transcript models were aligned to the assembled genomes of rheMacS and rheMac8 using GMAP[75], respectively. The number of mean consensus number per isoform and the collapse isoforms were counted after filtering by identity and coverage (default arguments: identity rate > 0.99 and coverage rate > 0.95) (Supplementary Table 11). In addition, to compare the annotation quality between rheMac8 and rheMacS, the RNA-seq short reads from 16 tissues were mapped to rheMac8 and rheMacS using Bowtie2 (v2.3.4.3)[76] and HISAT2[77], respectively (Supplementary Data 1).

**Repeat analysis**. Repetitive sequences including tandem repeats and transposable elements (TEs) were identified. First, we used Tandem Repeats Finder (TRF, v4.07b)[21] to annotate the tandem repeats with options "5 5 5 80 40 20 10 -m -ngs -h". Then, TEs were identified using RepeatMasker (v4.0.6)[20] to search against the known Repbase TE library (Repbase21.08).

**rheMacS SV detection**. For long-read PacBio data, we used the mapping software NGLMR[23] to align the subreads to the rhesus macaque reference genome rheMac8. Then Sniffles[23] was used to call SVs from the bam file. For the NGS short-read Illumina data, we mapped the reads to rheMac8 by using BWA. After sorting and removing duplicates, we used Delly[70] to call SVs. IGV (Integrative Genomics Viewer, http://software.broadinstitute.org/software/igv) was used for SV visualization.

The array-based and NGS-based SV sets of rhesus macaque were obtained from dbVar (https://www.ncbi.nlm.nih.gov/dbvar) and previous studies[24–26]. The SVs were lifted over to the rheMac8 coordinates using LiftOver tool, and only the lift-over SVs were included in downstream analyses. The same SV was defined with

overlapping length reaching at least 50% of the reciprocal similarities between published SVs and rheMacS SVs.

SV distribution were counts based on sliding windows (window size = 500 Kbp). Before plotting the distribution of bin counts with telomere distance, we identifies the centromeres of macaque in rheMac8 based on the results of[30], and the coordinates of each chromosomes' centromeres were lifted over from rheMac2 to rheMac8.

SVs genotyping was conducted with Illumina deep sequencing data (HiSeq X10; PE read, > 50-fold depth) in five unrelated Chinese rhesus monkeys using SVTyper[28]. As insertion is unsupported for SVTyper, we obtained insertion genotypes by mapping WGS reads to rheMacS, which containing the novel insertion sequences in assembly. According to genotyped as deletion in the rheMacS-based alignments, we inferred insertion genotypes of each individual.

**Identification of ASSVs.** Genome comparisons were performed using smartie-sv[7]. We mapped each long-read assembled ape genome to the rheMacS assembly, including human-ZF1 (ZF1), Chimpanzee-Clint_PTRv1 (CCP), Gorilla-Susie3 (GS3), and Orangutan-Susie_PABv1 (OSP) (Supplementary Table 16). Each chromosome of rheMacS was mapped to each ape genome sequence for SV calling, which was referred to as forward calling. For the reverse calling, we called the SVs by mapping each chromosome of each ape back to the rheMacS genome (Supplementary Fig. 23). Then, we filtered the SVs by intersecting two SV sets and obtained the high-confident SVs for each genome pair. ASSVs were identified by overlapping all rheMacS–ape SV sets (Supplementary Fig. 11).

Given the high-quality annotation of the reference human genome, we also mapped rheMacS to GRCh38 (V38) to call SVs. The called ASSV sets based on ZF1 and GRCh38 were overlapped to obtain the SVs with GRCh38 coordinates for better annotation (Supplementary Fig. 11). Furthermore, to exclude the monkey-lineage-specific SVs, we used the published marmoset genome (assembly name: ASM275486; assembly ID: GCA_002754865.1) (Supplementary Table 16).

**Annotation analysis for ASSVs.** ASSV annotation was performed using VEP with the GRCh38 coordinates, and the upstream and downstream distances were both 5 Kbp.

**Simulation analysis of ASSVs.** We generated a null distribution by randomly selecting a set of SVs as "ASSVs" from all rheMacS-GRCh38 SVs over many iterations (one million times). We then assessed those SVs against the genes they intersect to determine how often genes are associated with one of the four traits (tail development, microcephaly, body size, and adducted thumbs).

**Identification of ASSVs located in ADEs.** The published ChIP-seq data were used to identify ADEs in eight brain regions[46]. We downloaded the 51,283 genomic regions with H3K27Ac modification signals, which were generated from three humans, two chimpanzees and three rhesus monkeys in each of the eight brain regions, including PFC, precentral gyrus (PcGm), occipital pole (OP), caudate nucleus (CN), putamen (Put), CB, white matter (WM), and thalamic nuclei (TN). We pair-wisely compared the counts of each enhancer among three species (macaque–human, macaque–chimpanzee, and human–chimpanzee) and used the two-tailed unpaired $t$ test statistical assessment. We defined an enhancer as an ADE if this enhancer showed significant difference ($P < 0.05$) of the same direction in both macaque–human and macaque–chimpanzee comparisons while showing no difference ($P > 0.05$) in the human–chimpanzee comparison. According to this criterion, we obtained 7155 ADEs (Supplementary Data 7). To explore whether ASSVs occurred within these ADEs, we intersected the 7155 ADEs and our ASSV set, and we found 87 ADEs disrupted by 111 ASSVs near 65 genes. To quantify these ADEs, we calculated the $\log_2$(fold-change) between apes (average of human and chimpanzee) and rhesus macaque. Functional annotation clustering of ASSVs was performed using DAVID (v6.8)[78].

**Comparative gene expression analysis of neocortical layers.** The gene expression data of neocortical layers in human, chimpanzee, and macaque were obtained from the published data[48], which contains samples from four humans, four chimpanzees and four macaques of neocortical layers, including 16 laminar sections (section 1S–16S) and part of the adjacent white matter (section 17 S). The two-tailed paired $t$ test was used in accessing expression differences between apes and macaques.

**Identification of ASSVs related to ape-specific traits.** We collected the tail development-related genes set from three sources: (i) genes related to caudal development and defects from the literature; (ii) genes that showed tail-length changes (absent tail, short tail, and long tail) and caudal vertebral body fusion phenotype in the mouse gene knockout models from the MGI database (http://www.informatics.jax.org/); and (iii) genes with reportedly caudal regression syndrome or sacral agenesis in patients. An integrated table is shown in Supplementary Table 19. The genes causing primary microcephaly were obtained from the previous study[79]. The genes related to adducted thumbs were collected from Human Phenotype Ontology (https://hpo.jax.org/app/ ID: HP:0001181). The body

size-related genes were collected from the MGI database and literature that showed body size related changes (such as short stature and growth failure).

**ASSVs manual check and PCR validation.** The sequences of the ASSV regions were extracted by SAMtools "view" command, and sequence comparison was performed and plotted by MUMmer and LASTZ. The candidate ASSVs with 1 Kbp upstream/downstream sequences were aligned to the marmoset and gibbon genomes using MUMmer and blastn, and the best alignment score region was defined as the potential ASSV region in marmoset and gibbon. The ASSVs were classified into three categories based on manual check: (i) If the potential ASSV region and its 1 Kbp flanking sequences of marmoset could be completely aligned to the corresponding ASSV region of macaque using MUMmer and MEGA-X[80] (MUSCLE under default parameters), indicating that the regional sequences (SV ± 1 Kbp flanking) were orthologous between marmoset and macaque. These ASSVs were defined as "high-confident" ASSVs; (ii) If the potential ASSV region with 1 Kbp flanking sequences of marmoset could only be partially aligned to the corresponding ASSV sequence of macaque, we classified these ASSVs as "uncertain" ASSVs; and (iii) If the potential ASSV region with 1 Kbp flanking sequence of marmoset could be completely aligned to the corresponding ASSV sequence of macaque except for the SV regions, suggesting that these SVs also exist in marmoset, we then defined these ASSVs as "false" ASSVs. The same criterion was employed when using the gibbon genome to distinguish ASSVs from GASSVs. To further evaluate the identified 17,000 ASSVs, we mapped all the ASSVs with 1 Kbp upstream/downstream sequences of rheMacS to the NGS marmoset genome using MUMmer and blastn with default parameters, and the high alignment score region (with a stringent cutoff: blast: E-value = 0 and bit-score > 1000; MUMmer: Sequence identity > 85%) was defined as the potential ASSV regions in marmoset. ASSVs below the threshold were considered as uncertain ASSVs.

For PCR and Sanger sequencing validation, we included one rhesus macaque, one western hoolock gibbon (*Hoolock hoolock*), one chimpanzee and one human. The primers were designed by Primer Premier5, and the extended 100 bp sequences were included at the ASSV breakpoints as the PCR targets. The PCR products were visualized by agarose gel electrophoresis to verify the lengths of ASSVs and for Sanger sequencing. All the presented SVs in this study were validated both by PCR and Sanger sequencing. All primers used in this study were listed in Supplementary Data 10.

**Ethics statement.** The monkey was housed at the AAALAC (Association for Assessment and Accreditation of Laboratory Animal Care) accredited facility of Primate Research Center of Kunming Institute of Zoology. The animal protocol was approved in advance by the Institutional Animal Care and Use Committee of Kunming Institute of Zoology (Approval No: SYDW-2010002).

**Reporting summary.** Further information on research design is available in the Nature Research Reporting Summary linked to this article.

## Data availability
The PacBio sequence data, Illumina sequencing reads, rheMacS assembly, and its annotation files have been deposited in NCBI and GSA (Genome Sequence Archive) under the project accession numbers of PRJNA514196 and PRJCA001197, respectively. All source data underlying figures in this study are provided as a Source Data file. All other relevant data are available upon request.

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

## Acknowledgements

We thank T. Brown for assistance in editing this manuscript. This study was supported by grants from the Strategic Priority Research Program of the Chinese Academy of Sciences (XDB13010000), Ten thousand talents program of Yunnan province, and the National Natural Science Foundation of China (31730088 and 31621062). This work was supported, in part, by US National Institutes of Health (NIH) grant R01HG002385 to E.E.E. E.E.E. is an investigator of the Howard Hughes Medical Institute.

## Author contributions

B.S., X.Q., X.L., and Y.H. designed the project; X.L., T.H., X.M. collected the samples; Y.H., B.Z., J.J., X.M., Z.N.K., P.A., E.E.E., Y.G., and Y.Y. performed bioinformatics analyses; T.H., B.Z., and Y.H. performed genotyping and sequencing experiments; B.S. and Y.H. wrote the manuscript. All authors discussed the results and implications and commented on the manuscript.

## Additional information

**Competing interests:** Dr. Evan E. Eichler is on the scientific advisory board (SAB) of DNAnexus, Inc. The remaining authors declare no competing interests.

