## [Peer Review File · Nature Communications]

Reviewers' comments:

Reviewer #1 (Remarks to the Author):

This manuscript reports a new de novo PacBio assembly of the Chinese rhesus macaque genome. Rhesus macaques are among the most significant and widely used primates in biomedical research, and thus high-quality genomic information, including a high continuity reference genome assembly, is important. This new assembly is a substantial improvement over both the best available assembly for Chinese rhesus and the best available assembly for Indian rhesus macaques. The authors have used a strong combination of long-read sequencing, BioNano scaffolding data, short read Illumina data and Hi-C data to generate a high-quality reference genome assembly. In addition, the authors have generated long-read Iso-Seq RNA sequence and short read Illumina RNA sequence data for annotation of protein-coding and non-coding genes. The results of this work constitute a valuable contribution to primate genomics.

Given the continuity and completeness of the rhemaS assembly, the authors take advantage of the new opportunity to investigate rhesus macaque structural variation. By comparing Illumina short read sequences from 5 additional Chinese rhesus to the rhemaS reference, they identify 53,916 SVs larger than 50bp, most of which are polymorphic in macaques. These new data are not analyzed in great detail, but they begin to outline polymorphic structural variation in this important biomedical laboratory primate. This part of the paper is satisfactory, but does not provide much new biological insight.

The most interesting part of this work is the application of the new rhesus assembly to investigate ape-specific and great ape-specific structural variation. By applying the new Chinese rhesus macaque assembly to the question of ape-specific structural changes, the authors provide new insight into primate genomic evolution. The efforts to connect specific ASSVs to phenotypes generate mixed results. Some of the conclusions drawn by the authors are well-justified and make real contributions. The observations and conclusions regarding ITSN2 and NEDD9 seem to me to be reasonably well supported and valuable. But there are other examples described for which the evidence is much less persuasive and these cases should either be deleted or more supporting data should be included. I consider the examples of CDH8 and WDR62 in this later unpersuasive category. Do the H3K27c peaks and DNase signals in TRNP1 overlap with the newly described insertion, or are they simply all found in the same intron? If there is no overlap between the insertion and predicted regulatory sequences, then this case is another questionable one. The association of GASSVs in CPE, COL9A3 and ERCC5 should also be better justified and explained. Do the GASSVs map onto known regulatory elements, or simply fall into these genes? More detail is needed in these particular cases related to body size if the authors are going to provide adequate evidence to support their claims of likely functional effects.

The second paragraph of the Discussion is much too superficial. We already know with great certainty that coding sequences are under functional constraint, and that inter-genic sequences show more rapid evolution in terms of both single nucleotide substitutions and SVs. Stating that this study found 99% of SVs are outside coding regions, and that this illustrates something new, or that the difference in SVs between coding and non-coding regions demonstrates that non-coding regions "can provide more resources for phenotypic evolution" is not new. This conclusion is neither insightful or truly dependent on the new data presented in this paper. This paragraph needs thorough rethinking and re-writing.

Other concerns about the manuscript:

1) Figure 3: This figure apparently deals only with SVs larger than 1kb, but page 9 of the manuscript

defines SVs as changes larger than 50 bp. The authors need to clarify how these illustrations in Fig 3 correlate with the numbers cited in the text. For example, does the pie chart in Panel B show only SVs larger than 1kb, or all SVs larger than 50bp? Same for Panels C, D and E?

2) Figure 4: In my opinion, Panel B adds very little to the presentation. Why is it important to show the correspondence between ape and rhesus chromosomes, when none of the SVs are mapped onto those chromosomes individually?

3) Figure 4, Panel C: It looks like nearly all of these insertions and deletions are about 300bp. Are they primarily Alu insertions?

4) Page 19: Is "axiation" a word?

Reviewer #2 (Remarks to the Author):

In their manuscript "Long-read assembly of the Chinese rhesus macaque genome and identification of ape-specific structural variants", He and colleagues report a novel genomic resource for the rhesus macaque. Using a combination of long-read sequencing, optical mapping and Hi-C chromatin interaction mapping, they generate a de-novo assembly with greatly improved contiguity and completeness. Furthermore, they use long-read RNA sequencing to aid genome annotation. They identify structural variation via read mapping and comparing assemblies. These variants are then further refined into ape-specific variants using other genome assemblies from other monkeys, apes and humans.

Overall, I think this a well-executed study supported by a large amount of data and thus constitutes a valuable contribution to field. However, I found it – even for a genome assembly paper – overly descriptive with very few firm questions and subsequent analyses answering these. Especially the sections of SVs influencing gene expression and thus phenotypic traits often invoked vague statements lacking analytical support. I am aware that studies which employ novel technologies are often driven in this direction due to the lack of biological replicates or experiments. However, I am not sure if it is advisable to use some rather nebulous statements regarding the genetic basis of phenotypic traits, particularly when it comes to ape (and thus also human) evolution. More specifically, in the section on genes being influenced by SVs, I wondered how often one finds such a relationship just by chance. A simulation approach could be used to test for that.

Thus, I recommend that the authors address these points and revise the overly speculative sections to make this manuscript more suitable for the broad readership of Nature Communications.

Detailed review:

1. Page 3, end of first paragraph: 'With so many gaps, it has been difficult to systematically identify structural variants (SVs) in the macaque genome,...' The relationship between gaps and reliable SV calling might not be clear to the reader. Please support this statement with a reference and maybe a brief explanation.
2. Page 3, second paragraph, first sentence: The last part of the sentence (multiple scaffolding techs) does not make sense to me. These scaffolding approaches can also be used in combination with short read sequencing. Also, the last sentence of this paragraph is odd: Why is PacBio emphasized here, especially since the refs don't provide a comparison to other single-molecule technologies such as optical mapping. Please clarify.

3. Page 4, Results, line 5: 'contigs' instead of 'contig'.
4. Page 4, results, line 4: Is there a particular reason for why 'FALCON' instead of 'FALCON UNZIP' was used? The latter results in a diploid assembly and would thus potentially be more accurate. Given that this is a diploid organism, heterozygous SVs could result in chimeric / erroneous contigs, impeding their detection.
5. Page 5, last paragraph: '... that represents 2.95 Gb of the euchromatin ...' How was determined whether these 2.95 Gb of sequence belonged only to euchromatin? Please clarify.
6. Page 5, first paragraph, last sentence: The statement 'anchor scaffolds to chromosomes' might be more accurate by replacing 'chromosomes' with 'chromosome models'.
7. Figure 2B: The spikes in closed gap lengths are intriguing. Is there an explanation for these?
8. Page 8, first paragraph '... we also mapped 50-fold Illumina short read data ...': Is this the same data which was used in the Pilon polishing step? If so, isn't it surprising that there are any errors left?
9. Page 9, identification of SVs: There is no information on how SVs have been called (tools for SNV calling are mentioned). I suppose these were called via read mapping and Sniffles? Please clarify.
10. Page 10, last paragraph: Again, there is no information on how analyses (in this case the validation of SVs) were done.
11. Figure 3F: The depiction of variant validation with Sanger is very misleading / wrong. The variant is reportedly 4.3 kb of size, while the deletion in the sanger sequence is only a little over 100 bp. Please clarify.
12. Page 12, last paragraph: The sentence: 'Although no functional category reached statistical significance, we did observe several interesting functional categories among the top-ranked categories.' essentially contains no information. It is not clear whether 'top-ranked categories' are meaningful in a scientific sense. Please adjust accordingly. I also suggest replacing the term 'hypothesize' with 'speculate'.
13. Figure 4B: The synteny labelling is not clear to me. For example, if the color labels correspond to ape chromosomes, why is the second (dark green) macaque chromosome not 3 then?
14. Page 15, 'ASSVs in gene-coding regions', 3rd line '... and the other 7 were false (not ape-specific)': Does this mean that the variant was correctly called but was just not ape-specific? Or was it erroneously called in the first place?
15. Page 15, 'ASSVs in gene-coding regions': Please explain 'nonsense-mediated decay' and provide references.
16. Page 20, last paragraph, sentence 'In contrast, SVs in the noncoding regions (together with other genetic changes such as SNVs) are under much less constraint, and they can provide more resources for phenotypic evolution.': This sentence is not clear to me; how can SVs in noncoding regions both be under less constraint (i.e.; leading to fewer phenotypic changes) and provide more resources for phenotypic evolution (i.e.; leading to more phenotypic changes). Please clarify and also add references to support your statement.
17. Page 20, following the sentence above: Are 80 % of all SVs deletions? Or just the ape-specific ones? If the former, then this could also constitute a technical bias (deletions might be easier to discover) rather than a biological reason. Also, this seems a bit counterintuitive given that transposable element insertions are an important source of SVs. Please discuss these possibilities.
18. Page 21, first paragraph: Why were only insertions and deletions considered? Please clarify.
19. Page 22, second paragraph, PCR and Sanger sequencing of 25 coding ASSVs: To infer from these 25 SVs that the majority of the identified novel SVs are true is likely an overestimation, especially since SVs in coding vs. non-coding regions usually exhibit different features (e.g. SVs in tandem repeat arrays are more common in coding regions, but also are more likely false-positives). This should be taken into consideration.
20. Page 25, Gap closure: How many Ns in a run were treated as a gap?

Reviewer #3 (Remarks to the Author):

Summary: In this manuscript, He et al. present a long-read assembly of the Chinese rhesus macaque and use the new reference assembly as a basis for a comparative analysis of the structural variation present in resultant ape-lineage genomes. The quality of the reference genome assembly is quite good, and the data presented suggest that it will be useful to researchers involved in the study of the genetics of old-world monkeys. In order to demonstrate the utility of their reference in comparative genetic analysis, the authors use it to identify ape-specific structural variation (ASSV). I find that this analysis is far less refined, and the high false positive rates (~40% in interrogated coding regions) suggest limitations in their analysis. Given the lack of line numbers in the submission, my comments on the manuscript are listed in the order in which they are encountered in the text.

Page 8: The references to Du and Shi do not provide the methods for short-read alignment QV estimation. Are the authors referring to QV estimates similar to those from Bickhart et al. 2017 or Koren et al. 2018? Please provide details on this calculation and cite appropriate sources.

Page 10: The authors need to describe the CNV region detailed in figure 3F more carefully. From their own image subset, it appears that the read depth of WGS reads in the intron of ENSMMUG00000031115 is quite high and would result in a duplication instead of the deletion when read-depth CNV estimation tools are used. It would be important to note the exact details on why this is the case (ie. high prevalence of repeat elements in the intron, or alignment mistakes).

Page 12: Detection of ASSVs: How many ASSVs from the Marmoset - Macaque comparative alignments are likely due to alignment of short-read contigs to long-read contigs? The authors note the discrepancy in quality among the two reference genomes; however, it should be possible to list the number of "uncertain" ASSVs from this pairwise comparison in the text.

Mechanisms of ASSV formation: Are there any clues as to the origins of ASSVs in the assemblies studied? My first thoughts would be that repetitive elements are likely to intersect with a majority of identified ASSVs which would indicate potential Mobile-element mediated insertions/deletions. Additionally, NAHR could be detected by identifying expansion/compression of repetitive elements that intersect with ASSVs.

Figure 4A: The figure and figure caption do not clearly indicate all of the details in this subset. For example, are the numbers intersecting the right-most colored lines the number of forward- and reverse-mapped structural variants between each ape species and the Macaque?

Figure 4B: I do not understand the utility of this plot. The only novel information that this plot displays is the correspondence of Macaque chromosomes with ape-lineage chromosomes. The caption suggests that this plot shows that the orthologous locations of ASSVs validates chromosome synteny; however, it does not perform at this intended function. Obviously, plotting 17,000 purported ASSVs would be far too dense for a figure, but this information could be summarized far more succinctly. Perhaps the authors could plot the synteny of ASSVs on the only chromosomes that have discrepancies (chr 17 and 5; Gorilla translocation)?

SV breakpoint analysis figures: Figures that encompass the nucleotide sequence surrounding breakpoints in comparative alignment SVs are a good addition to the manuscript. My only concern is the use of poorly outlined hard-breaks in nucleotide sequence within the regions highlighted. For example, the TRNP1 alignment plot in figure 5E does not have a very clear "break" in nucleotide sequence contiguity. My recommendation would be to place a pair of black lines where the break in sequence is in the alignment. This way the reader will know where the alignments have been broken in order to highlight the breakpoint-flanking sequence.

PCR validation of coding ASSVs: While the authors report a high (64%) validation rate for coding ASSVs, aren't the majority of reported ASSVs within intergenic regions? If these ASSVs are present in repetitive sequence, can they be reliably confirmed via PCR and Sanger sequencing?

"Explore the ASSVs disrupted ADEs" Please revise this section header to conform to English grammar rules.

ASSVs manual check...: I am quite confused by the categories that the authors propose. According to the results and methods, weren't the ASSVs already filtered to remove "false" ASSV sites? Also, why wouldn't "uncertain" ASSVs simply be removed from consideration given their potential for being in the monkey lineage? Finally, this categorization is not described in the results section. I would recommend that the authors mention that they categorized ASSVs before mentioning "high quality" ASSV intersections with genes in the results.

Sincerely,
Derek M Bickhart

Reviewer #1:

This manuscript reports a new de novo PacBio assembly of the Chinese rhesus macaque genome. Rhesus macaques are among the most significant and widely used primates in biomedical research, and thus high-quality genomic information, including a high continuity reference genome assembly, is important. This new assembly is a substantial improvement over both the best available assembly for Chinese rhesus and the best available assembly for Indian rhesus macaques. The authors have used a strong combination of long-read sequencing, BioNano scaffolding data, short read Illumina data and Hi-C data to generate a high-quality reference genome assembly. In addition, the authors have generated long-read Iso-Seq RNA sequence and short read Illumina RNA sequence data for annotation of protein-coding and non-coding genes. The results of this work constitute a valuable contribution to primate genomics.

Given the continuity and completeness of the rhemaS assembly, the authors take advantage of the new opportunity to investigate rhesus macaque structural variation. By comparing Illumina short read sequences from 5 additional Chinese rhesus to the rhemaS reference, they identify 53,916 SVs larger than 50bp, most of which are polymorphic in macaques. These new data are not analyzed in great detail, but they begin to outline polymorphic structural variation in this important biomedical laboratory primate. This part of the paper is satisfactory, but does not provide much new biological insight.

1. The most interesting part of this work is the application of the new rhesus assembly to investigate ape-specific and great ape-specific structural variation. By applying the new Chinese rhesus macaque assembly to the question of ape-specific structural changes, the authors provide new insight into primate genomic evolution. The efforts to connect specific ASSVs to phenotypes generate mixed results. Some of the conclusions drawn by the authors are well-justified and make real contributions. The observations and conclusions regarding ITSN2 and NEDD9 seem to me to be reasonably well supported and valuable. But there are other examples described for which the evidence is much less persuasive and these cases should either be deleted or more supporting data should be included. I consider the examples of CDH8 and WDR62 in this later unpersuasive category. Do the H3K27c peaks and DNase signals in TRNP1 overlap with the newly described insertion, or are they simply all found in the same intron? If there is no overlap between the insertion and predicted regulatory sequences, then this case is another questionable one. The association of GASSVs in CPE, COL9A3 and ERCC5 should also be better justified and explained. Do the GASSVs map onto known regulatory elements, or simply fall into these genes? More detail is needed in these particular cases related to body size if the authors are going to provide adequate evidence to support their claims of likely functional effects.

Authors' response: For each presented case in the manuscript, we reassessed their

potential functional effects and the functions of the associated genes.

- 1) For the ASSVs in *ITSN2* and *NEDD9*, we further analyzed the published data of RNA expression in neocortical layers in human, chimpanzee and macaque (He, et al. 2017). We detected a significantly lower expression of *ITSN2* in apes (human and chimpanzee) than in macaque (Figure 4D, Figure S18), providing further support to the observed lower H3K27Ac signals of apes in both prefrontal cortex (PFC) and white matter (WM). The results suggest that the ASSV in *ITSN2* may reduce the enhancer activities and therefore lead to lower gene expression in apes. In the revised manuscript, we added a panel (Figure 4B) showing the decreased H3K27Ac signals in apes. We also updated Figure 4D to present the *ITSN2* expression divergence among humans, chimpanzees and macaques in the PFC cortical layers. For the ASSV (1,128 bp deletion) in *NEDD9*, humans and apes showed much higher H3K27Ac signals than macaques in cerebellum. However, gene expression data of cerebellum in nonhuman primates are not available to check its effect on *NEDD9* expression.
- 2) For the ASSV cases in *TRNP1* and *WDR62*, since current brain data (ChIP-seq and RNA-seq) (He, et al. 2017; Xu, et al. 2018) do not provide strong support for their functional effects, we removed these two cases in the revised manuscript.
- 3) For the ASSVs in genes associated with other ape-specific phenotypes (ASPs), i.e., taillessness (*MAP3K7* and *CDH8*) and body size (*CPE*, *COL9A3* and *ERCC5*), we did observe H3Ac27 and DNase signals (though relatively weak) in the ASSV regions based on the annotation information of the human genome from UCSC database. Regulatory elements such as enhancers usually function in a tissue-specific manner. Due to the lack of ENCODE data in the correspondent tissues of nonhuman primates, we cannot determine whether the observed weak signals in humans are caused by the disruption of strong regulatory elements in macaques (ancestral situation), or there are just no significant changes between apes and monkeys. In the revised manuscript, we downplayed these cases by pointing out that the functional roles of these ASSVs need to be further tested when the ENCODE data from nonhuman primates is available.
- 4) For the ASSVs in *NALCN*, a gene associated with adducted thumb, we identified four ASSVs. Though all four ASSVs were annotated as intronic variants by VEP, we found one of them (ASSV30349, Table S26) located in an ADE in the CB brain region (Xu, et al. 2018). Whether this ASSV causes regulatory changes in thumb development is yet to be tested when the nonhuman primate ENCODE data is available.

In summary, following the reviewer's comment and suggestion, we strengthened the cases of *ITSN2* and *NEDD9* with expression data. We removed the cases of *TRNP1* and *WDR62* due to the weak supporting data. We kept the cases demonstrating their potential roles in the emergences of the other ape-specific traits, including taillessness, body size and adducted thumb. These cases are promising candidates though the evidence is not strong due to the lack of ENCODE data of the correspondent tissues in nonhuman primates.

2. The second paragraph of the Discussion is much too superficial. We already know with great certainty that coding sequences are under functional constraint, and that inter-genic sequences show more rapid evolution in terms of both single nucleotide substitutions and SVs. Stating that this study found 99% of SVs are outside coding regions, and that this illustrates something new, or that the difference in SVs between coding and non-coding regions demonstrates that non-coding regions “can provide more resources for phenotypic evolution” is not new. This conclusion is neither insightful or truly dependent on the new data presented in this paper. This paragraph needs thorough rethinking and re-writing.

Authors’ response: Following the reviewer’s suggestion, we deleted the discussion on functional constraint. In the revised manuscripts, we rewrote this part focusing on the discussion of SV types and the potential molecular mechanism of SV origin.

Other concerns about the manuscript:

1) Figure 3: This figure apparently deals only with SVs larger than 1kb, but page 9 of the manuscript defines SVs as changes larger than 50 bp. The authors need to clarify how these illustrations in Fig 3 correlate with the numbers cited in the text. For example, does the pie chart in Panel B show only SVs larger than 1kb, or all SVs larger than 50bp? Same for Panels C, D and E?

Authors’ response: We defined SVs as sequence change ≥ 50 bp, and all downstream analysis and descriptions were based on this SV set. However, there are too many SVs in each chromosome to have a clear presentation (as shown in Figure R1), so we only displayed the SVs ≥ 1 Kbp in the distribution plot (Figure 3A). For clarity, we added a better description to the figure legend.

Figure R1. The distribution of 53,916 SVs (≥ 50 bp in size) among the rhesus macaque chromosomes.

2) Figure 4: In my opinion, Panel B adds very little to the presentation. Why is it

important to show the correspondence between ape and rhesus chromosomes, when none of the SVs are mapped onto those chromosomes individually?

Authors' response: Following the reviewer's suggestion, we moved Panel B to the supplementary data.

3) Figure 4, Panel C: It looks like nearly all of these insertions and deletions are about 300bp. Are they primarily Alu insertions?

Authors' response: We annotated repeats for all 17,000 identified ASSVs using RepeatMasker. As shown in Table R1 (added as Supplementary Table 21 in the revised version), among these ASSVs, 75.47% insertions and 80.62% deletions are repeats; 21.04% insertions and 40.01% deletions are Alus.

Table R1. Repeat annotation of 17,000 ASSVs.

	Insertions			Deletions			
ASSV number	3544			13456			
Total bases	2139296			5830871			
Marked bases	1614579			4700799			
Percentage (%)	75.47			80.62			
	number	length(bp)	percentage (%)	number	length(bp)	percentage (%)	
SINEs:	2126	470389	21.99	10074	2366221	40.58	
ALUs	1966	450009	21.04	9813	2332824	40.01	
MIRs	160	20380	0.95	260	33312	0.57	
LINEs:	1080	677585	31.67	2017	1534500	26.32	
LINE1	933	645236	30.16	1800	1493075	25.61	
LINE2	136	30030	1.4	200	39329	0.67	
L3/CR1	9	1493	0.07	12	1651	0.03	
LTR	elements:	578	386355	18.06	1056	577699	9.91
ERV_L	73	25585	1.2	137	49418	0.85	
ERV_L-MaLRs	195	63345	2.96	301	76188	1.31	
ERV_classI	245	238640	11.16	425	290148	4.98	
ERV_classII	57	57095	2.67	180	159179	2.73	
DNA	elements:	219	42877	2	311	55328	0.95
hAT-Charlie	105	18104	0.85	155	21965	0.38	
TcMar-Tigger	54	13260	0.62	91	20843	0.36	
Unclassified	9	3064	0.14	1	467	0.01	
Small RNA	19	1483	0.07	32	3504	0.06	
Satellites	18	1486	0.07	2	530	0.01	
Simple repeat	483	28221	1.32	2169	145832	2.5	
Low complexity	45	3119	0.15	172	16771	0.29	

4) Page 19: Is “axiation” a word?

Authors’ response: Yes, the definition of axiation is the development of polarity in an embryo or its parts. In the revised manuscript, we replaced “axiation” with “body plan” as it is a more commonly used term.

Reviewer #2:

In their manuscript “Long-read assembly of the Chinese rhesus macaque genome and identification of ape-specific structural variants”, He and colleagues report a novel genomic resource for the rhesus macaque. Using a combination of long-read sequencing, optical mapping and Hi-C chromatin interaction mapping, they generate a de-novo assembly with greatly improved contiguity and completeness. Furthermore, they use long-read RNA sequencing to aid genome annotation. They identify structural variation via read mapping and comparing assemblies. These variants are then further refined into ape-specific variants using other genome assemblies from other monkeys, apes and humans.

Overall, I think this a well-executed study supported by a large amount of data and thus constitutes a valuable contribution to field. However, I found it – even for a genome assembly paper – overly descriptive with very few firm questions and subsequent analyses answering these. Especially the sections of SVs influencing gene expression and thus phenotypic traits often invoked vague statements lacking analytical support. I am aware that studies which employ novel technologies are often driven in this direction due to the lack of biological replicates or experiments. However, I am not sure if it is advisable to use some rather nebulous statements regarding the genetic basis of phenotypic traits, particularly when it comes to ape (and thus also human) evolution. More specifically, in the section on genes being influenced by SVs, I wondered how often one finds such a relationship just by chance. A simulation approach could be used to test for that.

Authors’ response: To address the question raised by the reviewer, we performed a simulation to test the randomization of ASSVs for each phenotypic trait. We generated a null distribution by randomly selecting a set of SVs as "ASSVs" from all rheMacS-GRCh38 SVs over many iterations (1 million times). We then assessed those SVs against the genes they intersect to determine how often genes are associated with one of the four traits (tail development, microcephaly, body size and adducted thumbs). The results showed significance for body size ($p=6.0E-06$) and microcephaly ($p=3.81E-04$) but not adducted thumbs ($8.8E-01$) or tail development ($p=8.99E-01$). This result suggests that there is enrichment of ASSVs associated with ape- or great ape-specific phenotypes, such as body size and microcephaly, and we have added the method description and results in the revised manuscript.

Thus, I recommend that the authors address these points and revise the overly speculative sections to make this manuscript more suitable for the broad readership of Nature Communications.

Detailed review:

1. Page 3, end of first paragraph: ‘With so many gaps, it has been difficult to systematically identify structural variants (SVs) in the macaque genome,..’ The relationship between gaps and reliable SV calling might not be clear to the reader. Please support this statement with a reference and maybe a brief explanation.

Authors’ response: Following the reviewer’s comment, we rephrased the sentence. Basically, the poor assembly quality of the published draft genome of Chinese rhesus macaque (Yan, et al. 2011) made it difficult to systematically identify SVs because of its poor contiguity (fragmentation) and incompleteness (many gaps). The misassemblies and missing regions of an NGS-based assembly most often stem from its limitation for assembly of repetitive DNA, including retrotransposons and segmental duplications (Alkan, et al. 2011). We have added a brief explanation and the relevant reference (Alkan, et al. 2011) in the revised text.

2. Page 3, second paragraph, first sentence: The last part of the sentence (multiple scaffolding techs) does not make sense to me. These scaffolding approaches can also be used in combination with short read sequencing. Also, the last sentence of this paragraph is odd: Why is PacBio emphasized here, especially since the refs don’t provide a comparison to other single-molecule technologies such as optical mapping. Please clarify.

Authors’ response: To be clear, we rephrased the sentence by stating the advantage of combining long-read sequencing and multiplatform scaffoldings.

3. Page 4, Results, line 5: ‘contigs’ instead of ‘contig’.

Authors’ response: This has been corrected accordingly.

4. Page 4, results, line 4: Is there a particular reason for why ‘FALCON’ instead of ‘FALCON UNZIP’ was used? The latter results in a diploid assembly and would thus potentially be more accurate. Given that this is a diploid organism, heterozygous SVs could result in chimeric / erroneous contigs, impeding their detection.

Authors’ response: Mapping short reads to a FALCON UNZIP assembly fails because there are two representations of many loci, and short-read mappers do not handle alternate loci well. For example, this complicates polishing procedures using Pilon, which is absolutely essential for quality. With another sample, we have been

experimenting with polishing UNZIP assemblies using Pilon and an ALT-aware alignment approach, but at this time, we cannot get Pilon to properly polish the alternate contigs. We acknowledge that haplotype-aware assemblies are becoming the standard; however, we have a FALCON assembly completed with current standards, and we have supported many of our findings over several primate species.

5. Page 5, last paragraph: ‘... that represents 2.95 Gb of the euchromatin ...’ How was determined whether these 2.95 Gb of sequence belonged only to euchromatin? Please clarify.

Authors’ response: To be accurate, we replaced “euchromatin” with “chromosomes”.

6. Page 5, first paragraph, last sentence: The statement ‘anchor scaffolds to chromosomes’ might be more accurate by replacing ‘chromosomes’ with ‘chromosome models’.

Authors’ response: As suggested by the reviewer, we have replaced “chromosome” with “chromosome models”.

7. Figure 2B: The spikes in closed gap lengths are intriguing. Is there an explanation for these?

Authors’ response: In primate genomes, the distribution of gaps in assemblies is not random (Figure R2) and their sizes are often represented as approximations because they are not accurately known. In GRCh38, for example, there are often gaps of 10,000 bp ($n = 48$), 50,000 bp ($n = 169$) and 100,000 bp ($n = 20$). Gaps of these three sizes alone account for 42% of all gaps in GRCh38 (237 of 559). The same pattern can be observed in rheMac8, except for small gaps (<100 bp in size), and there are often gaps in 100 bp, 100-1000 bp and 50,000 bp. These modal features are likely due to partially closed gaps.

Figure R2. The distribution of gap size in GRCh38 and rheMac8.

8. Page 8, first paragraph ‘..., we also mapped 50-fold Illumina short read data ...’: Is this the same data which was used in the Pilon polishing step? If so, isn’t it surprising that there are any errors left?

Authors’ response: Even with the best assemblies and polishing strategies, long-read assemblies still contain indel errors (Watson et al. 2019; Koren et al. 2019). The newest Sequel II HiFi protocol, which produces accurate long reads of QV 20+, cannot remove all of these assembly errors (Vollger et al. 2019). Although the error model of PacBio sequencing is random, homopolymer indels account for most of the remaining errors because of uncertainty introduced by random errors overlapped in homopolymer runs (e.g., "AAAAA..."). Pilon is also not effective in regions where short reads do not map uniquely, such as segmental duplications, regardless of sequencing depth or the number of polishing rounds. Although improvements in sequencing technology will ultimately remove many of these errors, this assembly represents current best practices.

9. Page 9, identification of SVs: There is no information on how SVs have been called (tools for SNV calling are mentioned). I suppose these were called via read mapping and Sniffles? Please clarify.

Authors’ response: Variants were called from aligned contigs using a method developed for this purpose (Chaisson, et al. 2015). Briefly, the alignment CIGAR string for each contig is traversed, nearby small insertions and deletions are merged, and the resulting reference or contig gap is output as an insertion or a deletion (respectively).

10. Page 10, last paragraph: Again, there is no information on how analyses (in this case the validation of SVs) were done.

Authors’ response: Genotyping of the 53,916 SVs in rheMacS was conducted with Illumina deep sequencing data (Hiseq X10; PE read, >50-fold depth) in five unrelated Chinese rhesus monkeys using SVTyper (Chiang, et al. 2015). By inputting breakpoints of SVs, the WGS data was mapped to rheMac8, and then SVTyper infers genotypes at each site based on a Bayesian algorithm. Because SVTyper cannot genotype insertions directly, we obtained insertion calls by mapping the WGS reads to rheMacS, and the called deletions were regarded as insertions. According to the genotyping results, we counted how many SVs could be genotyped (validated) by the NGS data. We provided brief descriptions in the revised text, and a full description is included in Methods.

11. Figure 3F: The depiction of variant validation with Sanger is very misleading / wrong. The variant is reportedly 4.3 kb of size, while the deletion in the sanger sequence is only a little over 100 bp. Please clarify.

Authors' response: In Figure 3F, we only showed the sequences (100 bp flanking sequences) surrounding the breakpoints of the deletion, and this was done by Sanger sequencing. However, as suggested by Reviewer #3, this case is not a good one to show the advantage of calling SVs using long-read sequencing because with proper calling tools, the short-read NGS data may also be able to call this SV. We therefore removed this case in the revised manuscript.

12. Page 12, last paragraph: The sentence: 'Although no functional category reached statistical significance, we did observe several interesting functional categories among the top-ranked categories.' essentially contains no information. It is not clear whether 'top-ranked categories' are meaningful in a scientific sense. Please adjust accordingly. I also suggest replacing the term 'hypothesize' with 'speculate'.

Authors' response: In the revised manuscript, we updated the phrase to "top 5% (10/208) categories". We also used the term "speculate" to replace "hypothesize" as suggested by reviewer.

13. Figure 4B: The synteny labelling is not clear to me. For example, if the color labels correspond to ape chromosomes, why is the second (dark green) macaque chromosome not 3 then?

Authors' response: The chromosome numbers are different among different primate species. Chromosome-2 of macaque is homologous to Chromosome-3 of apes and human (dark green) (Rogers, et al. 2006). As suggested by Reviewers #1 and #3, we moved it to supplementary materials.

14. Page 15, 'ASSVS in gene-coding regions', 3rd line '... and the other 7 were false (not ape-specific)': Does this mean that the variant was correctly called but was just not ape-specific? Or was it erroneously called in the first place?

Authors' response: All SVs mentioned in the ASSV section are correctly called—the "true" or "false" refers to whether they are ape specific. To be clear, we have added more explanation in the revised manuscript.

15. Page 15, 'ASSVs in gene-coding regions': Please explain 'nonsense-mediated decay' and provide references.

Authors' response: Nonsense-mediated decay (NMD) is a surveillance pathway that exists in all eukaryotes. Its main function is to reduce errors in gene expression by eliminating mRNA transcripts that contain premature stop codons (Baker and Parker 2004). Translation of these aberrant mRNAs could, in some cases, lead to deleterious gain-of-function or dominant-negative activity of the resulting proteins (Chang, et al. 2007). Given some NMD transcripts can be translated to functional proteins, VEP annotates those SVs located in the NMD transcripts as NMD transcript variants and

coding sequence variants. We added references in the revised manuscript.

16. Page 20, last paragraph, sentence ‘In contrast, SVs in the noncoding regions (together with other genetic changes such as SNVs) are under much less constraint, and they can provide more resources for phenotypic evolution.’: This sentence is not clear to me; how can SVs in noncoding regions both be under less constraint (i.e.; leading to fewer phenotypic changes) and provide more resources for phenotypic evolution (i.e.; leading to more phenotypic changes). Please clarify and also add references to support your statement.

Authors’ response: Variants in the noncoding regions are under much less constraint than coding regions. Less constraint means it is easier to accumulate mutations than coding regions under natural selection, and therefore would provide more resources for phenotypic evolution because they are unlikely to be immediately eliminated from the population (Bird, et al. 2007). However, as Reviewer #1 suggested, this discussion is not new and not very helpful for presenting our results, so we removed these sentences in the revised manuscript and rewrote this part.

17. Page 20, following the sentence above: Are 80 % of all SVs deletions? Or just the ape-specific ones? If the former, then this could also constitute a technical bias (deletions might be easier to discover) rather than a biological reason. Also, this seems a bit counterintuitive given that transposable element insertions are an important source of SVs. Please discuss these possibilities.

Authors’ response: The 80% ASSVs are deletions (Figure 4B in revised text) not all SVs. This was changed accordingly in the revised manuscript.

18. Page 21, first paragraph: Why were only insertions and deletions considered? Please clarify.

Authors’ response: There were two reasons for considering only insertions and deletions. First, they account for more than 98% SVs in the assemblies, and smartie-sv is good for calling these two types. Second, because we used the NGS-based genome data of gibbon and marmoset to identify SVs occurred in the ape lineage, i.e., the ASSVs, it would introduce ambiguity in calling the other SV types such as duplications and translocations. Indeed, these SV types may also contribute to phenotypic evolution in primates, and they are worth studying in the future.

19. Page 22, second paragraph, PCR and Sanger sequencing of 25 coding ASSVs: To infer from these 25 SVs that the majority of the identified novel SVs are true is likely an overestimation, especially since SVs in coding vs. non-coding regions usually exhibit different features (e.g. SVs in tandem repeat arrays are more common in coding regions, but also are more likely false-positives). This should be taken into consideration.

Authors' response: We agree with this point. In the revised manuscript, we rephrased the sentence to avoid overestimation.

20. Page 25, Gap closure: How many Ns in a run were treated as a gap?

Authors' response: A region consisting of continuous runs of Ns ($N>1$) in the rheMac8 chromosomes was defined as a gap. We have added this information in the revised text.

Reviewer #3:

Summary: In this manuscript, He et al. present a long-read assembly of the Chinese rhesus macaque and use the new reference assembly as a basis for a comparative analysis of the structural variation present in resultant ape-lineage genomes. The quality of the reference genome assembly is quite good, and the data presented suggest that it will be useful to researchers involved in the study of the genetics of old-world monkeys. In order to demonstrate the utility of their reference in comparative genetic analysis, the authors use it to identify ape-specific structural variation (ASSV). I find that this analysis is far less refined, and the high false positive rates (~40% in interrogated coding regions) suggest limitations in their analysis. Given the lack of line numbers in the submission, my comments on the manuscript are listed in the order in which they are encountered in the text.

1. Page 8: The references to Du and Shi do not provide the methods for short-read alignment QV estimation. Are the authors referring to QV estimates similar to those from Bickhart et al. 2017 or Koren et al. 2018? Please provide details on this calculation and cite appropriate sources.

Authors' response: We estimated QV (quality value) following the previous methods (Bickhart, et al. 2017; Koren, et al. 2018), mapping 50× depth Illumina data with BWA-MEM and identifying variants with GATK. We updated the references and provided details of the calculation in the Methods.

2. Page 10: The authors need to describe the CNV region detailed in figure 3F more carefully. From their own image subset, it appears that the read depth of WGS reads in the intron of ENSMMUG00000031115 is quite high and would result in a duplication instead of the deletion when read-depth CNV estimation tools are used. It would be important to note the exact details on why this is the case (ie. high prevalence of repeat elements in the intron, or alignment mistakes).

Authors' response: The deletion portion indeed shows increased read-depth based on NGS data, suggesting a common repeat or a duplication. As we explained above, this is

not a good case to show the advantage of long-read data; we therefore removed this case in the revised manuscript.

3. Page 12: Detection of ASSVs: How many ASSVs from the Marmoset - Macaque comparative alignments are likely due to alignment of short-read contigs to long-read contigs? The authors note the discrepancy in quality among the two reference genomes; however, it should be possible to list the number of “uncertain” ASSVs from this pairwise comparison in the text.

Authors’ response: We evaluated the ASSV situation of marmoset using blastn and MUMmer. Each ASSV with 1 Kbp upstream/downstream sequences of rheMacS were aligned to the NGS marmoset genome using MUMmer and blastn with default parameters, and the high alignment score region (with a stringent cut off: blastn: E-value=0 and bit-score>1000; MUMmer: Sequence-identity>85%) was defined as the potential ASSV regions in marmoset. Among the 17,000 identified ASSV, 9,266 ASSVs (54.51%) were below the threshold, meaning that poor sequence similarity between NGS-marmoset and TGS-macaque. We marked these as “uncertain” ASSVs in the revised manuscript (Table S20).

4. Mechanisms of ASSV formation: Are there any clues as to the origins of ASSVs in the assemblies studied? My first thoughts would be that repetitive elements are likely to intersect with a majority of identified ASSVs which would indicate potential Mobile-element mediated insertions/deletions. Additionally, NAHR could be detected by identifying expansion/compression of repetitive elements that intersect with ASSVs.

Authors’ response: As shown in Table R1, we annotated the repeat content of all ASSVs and found 78.05% map to common repeats in the genome such as SINEs, LINEs and LTRs. We added this result to the revised manuscript based on a previously described method (Audano, et al. 2019). We find that 30 ASSVs consist of tandem repeats and likely are the result of non-allelic homologous recombination (NAHR) although further work will be required to delineate mutational mechanisms. While this may be an interesting point, it will take method development and many hours to analyze because the signatures of the forces that shaped these events are likely obscured because of the evolutionary timescale (about 25 million years) involved.

5. Figure 4A: The figure and figure caption do not clearly indicate all of the details in this subset. For example, are the numbers intersecting the right-most colored lines the number of forward- and reverse-mapped structural variants between each ape species and the Macaque?

Authors’ response: The numbers on the colored line indicate the SV numbers by genome-pairwise comparisons. The numbers in the blue boxes represent the overlap for different pairwise results. For example, 134,820 refers to the SVs shared between

macaque-human and macaque-chimpanzee. The number in the red box (17,000) refers to the shared SVs among macaque-apes after excluding the macaque-lineage-specific SVs using marmoset as outgroup, i.e., the ape-specific SVs (ASSVs). We added more detailed descriptions to the figure legend for this panel.

6. Figure 4B: I do not understand the utility of this plot. The only novel information that this plot displays is the correspondence of Macaque chromosomes with ape-lineage chromosomes. The caption suggests that this plot shows that the orthologous locations of ASSVs validates chromosome synteny; however, it does not perform at this intended function. Obviously, plotting 17,000 purported ASSVs would be far too dense for a figure, but this information could be summarized far more succinctly. Perhaps the authors could plot the synteny of ASSVs on the only chromosomes that have discrepancies (chr 17 and 5; Gorilla translocation)?

Authors' response: This was also mentioned by Reviewer #1, and we moved it to supplementary materials.

7. SV breakpoint analysis figures: Figures that encompass the nucleotide sequence surrounding breakpoints in comparative alignment SVs are a good addition to the manuscript. My only concern is the use of poorly outlined hard-breaks in nucleotide sequence within the regions highlighted. For example, the TRNP1 alignment plot in figure 5E does not have a very clear “break” in nucleotide sequence contiguity. My recommendation would be to place a pair of black lines where the break in sequence is in the alignment. This way the reader will know where the alignments have been broken in order to highlight the breakpoint-flanking sequence.

Authors' response: It is a resolution problem. We fixed it in the revised version.

8. PCR validation of coding ASSVs: While the authors report a high (64%) validation rate for coding ASSVs, aren't the majority of reported ASSVs within intergenic regions? If these ASSVs are present in repetitive sequence, can they be reliably confirmed via PCR and Sanger sequencing?

Authors' response: Indeed, ~78% of ASSVs fall in or contain repeats (Table R1). By mapping each sequence from Sanger sequencing (~100 bp) to corresponding species genomes, we assessed the repeat content of the 47 tested ASSVs. Among the 25 coding ASSVs, all of them can be clearly evaluated since they either do not contain repeats or the Sanger sequence traverses a unique/repeat boundary. For the other 22 noncoding ASSVs, we found that two of them (ASSV34579 and ASSV34877) are fully repetitive (100% sequence identity) and therefore cannot be reliably confirmed by the Sanger sequences. These were excluded from downstream analysis. The remaining 20 noncoding ASSVs were all reliably validated. Of note, all the presented ASSV cases in the manuscript were PCR/Sanger validated.

9. “Explore the ASSVs disrupted ADEs” Please revise this section header to conform to English grammar rules.

Authors’ response: We rephrased the section header as “Identification of ASSVs located in ADEs” in the revised manuscript.

10. ASSVs manual check...: I am quite confused by the categories that the authors propose. According to the results and methods, weren’t the ASSVs already filtered to remove “false” ASSV sites? Also, why wouldn’t “uncertain” ASSVs simply be removed from consideration given their potential for being in the monkey lineage? Finally, this categorization is not described in the results section. I would recommend that the authors mention that they categorized ASSVs before mentioning “high quality” ASSV intersections with genes in the results.

Authors’ response: Although the final candidate 17,000 ASSVs were filtered by the NGS-marmoset genome, it is still hard to completely exclude all the macaque-lineage-specific SVs. The main reason is the poor quality of the marmoset genome, which led to poor alignments in some genomic regions. One way to identify the true ASSVs from these candidates is a manual check, i.e., a local sequence alignment of each SV and PCR validation. Considering it is impractical to manually check all 17,000 candidates, we focused on the candidate SVs located in genes of interest. We understand the reviewer’s concern, and we updated the text to clarify this point. When a high-quality genome of marmoset is available in the future, we will be able to pinpoint all the ASSVs with greater confidence. At present, this set should be regarded as candidate set for future evaluation.

Reference

- Alkan, C., S. Sajjadian, and E. E. Eichler Limitations of next-generation genome sequence assembly. *Nature Methods*. 8(1):61-65 (2011).
- Audano, P. A., et al. Characterizing the Major Structural Variant Alleles of the Human Genome. *Cell*. 176(3):663-+ (2019).
- Baker, K. E., and R. Parker Nonsense-mediated mRNA decay: terminating erroneous gene expression. *Current Opinion in Cell Biology*. 16(3):293-299 (2004).
- Bickhart, D. M., et al. Single-molecule sequencing and chromatin conformation capture enable de novo reference assembly of the domestic goat genome. *Nat Genet*. 49(4):643-650 (2017).
- Bird, C. P., et al. Fast-evolving noncoding sequences in the human genome. *Genome Biology*. 8(6) (2007).
- Chaisson, M. J., et al. Resolving the complexity of the human genome using single-molecule sequencing. *Nature*. 517(7536):608-11 (2015).
- Chang, Y. F., J. S. Imam, and M. E. Wilkinson The nonsense-mediated decay RNA surveillance pathway. *Annual Review of Biochemistry*. 76:51-74 (2007).
- Chiang, C., et al. SpeedSeq: ultra-fast personal genome analysis and interpretation. *Nature Methods*. 12(10):966-968 (2015).
- He, Z. S., et al. Comprehensive transcriptome analysis of neocortical layers in humans, chimpanzees

- and macaques. *Nature Neuroscience*. 20(6):886-+ (2017).
- Koren, S., et al. De novo assembly of haplotype-resolved genomes with trio binning. *Nat Biotechnol.* (2018).
- Koren, S., Phillippy, A. M., Simpson, J. T., Loman, N. J., & Loose, M. (2019). Reply to ‘Errors in long-read assemblies can critically affect protein prediction.’ *Nature Biotechnology*. <https://doi.org/10.1038/s41587-018-0005-y>
- Rogers, J., et al. An initial genetic linkage map of the rhesus macaque (*Macaca mulatta*) genome using human microsatellite loci. *Genomics*. 87(1):30-38 (2006).
- Vollger 2019: <https://www.biorxiv.org/content/10.1101/635037v2.full>
- Watson, M., & Warr, A. (2019). Errors in long-read assemblies can critically affect protein prediction. *Nature Biotechnology*, 1–3. <https://doi.org/10.1038/s41587-018-0004-z>
- Xu, C., et al. Human-specific features of spatial gene expression and regulation in eight brain regions. *Genome Research*. 28(8):1097-1110 (2018).
- Yan, G. M., et al. Genome sequencing and comparison of two nonhuman primate animal models, the cynomolgus and Chinese rhesus macaques. *Nature Biotechnology*. 29(11):1019-U89 (2011).

Reviewers' comments:

Reviewer #1 (Remarks to the Author):

This is a revised manuscript. In my opinion, the authors have adequately addressed the concerns and questions raised in the first round of review. I recommend publication of this revised paper.

Reviewer #3 (Remarks to the Author):

Summary: In this revision, all of my major concerns were addressed by the authors. I still have several concerns with the discussion of the data that require revision, but I have more confidence in the results of the ASSV analysis.

On Tandem Repeats: The authors identified 30 candidate tandem repeat expansions of ASSVs but have not reported these in the manuscript. I understand that the determination of the exact mechanisms of SV formation require dedicated effort; however, I am satisfied if the authors merely report that 30 ASSVs appear to consist of tandem repeats but further efforts are needed to validate them.

Line 416: The authors state in their response that they also used Sanger sequencing to validate 20 out of 22 non-coding ASSVs (with 2 being difficult to confirm due to their completely repetitive nature). Despite this being a good validation of their dataset, I do not see reference to this in the text. Such descriptions do not need to be lengthy, but it would be important to make clearer reference to this in the text, particularly since non-coding regions are predicted to consist of the majority of SVs in multi-cellular Eukaryotes.

Line 456: As written, this sentence is confusing to the reader. I would prefer to see an addition similar to the authors' response to my point regarding the comparison of short-read to long-read assemblies here. I would be satisfied if the authors note this point as a potential discrepancy in their comparisons and that a long-read assembly for marmoset is needed for more accurate comparisons.

Reviewer #4 (who co-reviewed the first version of the manuscript with Reviewer 2) (Remarks to the Author):

Reviewer #2:

Reply:

I think the authors have addressed most of my comments nicely and overall improved the manuscript. Some of them however still need clarification in my opinion. Those comments will have a 'Reply:' section below the author's response.

In their manuscript "Long-read assembly of the Chinese rhesus macaque genome and identification of ape-specific structural variants", He and colleagues report a novel genomic resource for the rhesus macaque. Using a combination of long-read sequencing, optical mapping and Hi-C chromatin interaction mapping, they generate a de-novo assembly with greatly improved contiguity and completeness. Furthermore, they use long-read RNA sequencing to aid genome annotation. They

identify structural variation via read mapping and comparing assemblies. These variants are then further refined into ape-specific variants using other genome assemblies from other monkeys, apes and humans.

Overall, I think this a well-executed study supported by a large amount of data and thus constitutes a valuable contribution to field. However, I found it – even for a genome assembly paper – overly descriptive with very few firm questions and subsequent analyses answering these. Especially the sections of SVs influencing gene expression and thus phenotypic traits often invoked vague statements lacking analytical support. I am aware that studies which employ novel technologies are often driven in this direction due to the lack of biological replicates or experiments. However, I am not sure if it is advisable to use some rather nebulous statements regarding the genetic basis of phenotypic traits, particularly when it comes to ape (and thus also human) evolution. More specifically, in the section on genes being influenced by SVs, I wondered how often one finds such a relationship just by chance. A simulation approach could be used to test for that.

Authors' response: To address the question raised by the reviewer, we performed a simulation to test the randomization of ASSVs for each phenotypic trait. We generated a null distribution by randomly selecting a set of SVs as "ASSVs" from all rheMacS-GRCh38 SVs over many iterations (1 million times). We then assessed those SVs against the genes they intersect to determine how often genes are associated with one of the four traits (tail development, microcephaly, body size and adducted thumbs). The results showed significance for body size ($p=6.0E-06$) and microcephaly ($p=3.81E-04$) but not adducted thumbs ($8.8E-01$) or tail development ($p=8.99E-01$). This result suggests that there is enrichment of ASSVs associated with ape- or great ape-specific phenotypes, such as body size and microcephaly, and we have added the method description and results in the revised manuscript.

Reply:

Thank you, that certainly improves your point.

Thus, I recommend that the authors address these points and revise the overly speculative sections to make this manuscript more suitable for the broad readership of Nature Communications.

Detailed review:

1. Page 3, end of first paragraph: 'With so many gaps, it has been difficult to systematically identify structural variants (SVs) in the macaque genome,..' The relationship between gaps and reliable SV calling might not be clear to the reader. Please support this statement with a reference and maybe a brief explanation.

Authors' response: Following the reviewer's comment, we rephrased the sentence. Basically, the poor assembly quality of the published draft genome of Chinese rhesus macaque (Yan, et al. 2011) made it difficult to systematically identify SVs because of its poor contiguity (fragmentation) and incompleteness (many gaps). The misassemblies and missing regions of an NGS-based assembly most often stem from its limitation for assembly of repetitive DNA, including retrotransposons and segmental duplications (Alkan, et al. 2011). We have added a brief explanation and the relevant reference (Alkan, et al. 2011) in the revised text.

Reply:

The relationship between poor contiguity (many gaps) and SV calling is still not explained. To my knowledge, the problem lies in the fact that if an SV breakpoint lies in an assembly gap, it cannot be detected (correctly). Additionally, assembly gaps are often caused by repetitive DNA, which in turn

promotes the formation of SV. Maybe you can change the sentence accordingly.

2. Page 3, second paragraph, first sentence: The last part of the sentence (multiple scaffolding techs) does not make sense to me. These scaffolding approaches can also be used in combination with short read sequencing. Also, the last sentence of this paragraph is odd: Why is PacBio emphasized here, especially since the refs don't provide a comparison to other single-molecule technologies such as optical mapping. Please clarify.

Authors' response: To be clear, we rephrased the sentence by stating the advantage of combining long-read sequencing and multiplatform scaffoldings.

3. Page 4, Results, line 5: 'contigs' instead of 'contig'.

Authors' response: This has been corrected accordingly.

4. Page 4, results, line 4: Is there a particular reason for why 'FALCON' instead of 'FALCON UNZIP' was used? The latter results in a diploid assembly and would thus potentially be more accurate. Given that this is a diploid organism, heterozygous SVs could result in chimeric / erroneous contigs, impeding their detection.

Authors' response: Mapping short reads to a FALCON UNZIP assembly fails because there are two representations of many loci, and short-read mappers do not handle alternate loci well. For example, this complicates polishing procedures using Pilon, which is absolutely essential for quality. With another sample, we have been experimenting with polishing UNZIP assemblies using Pilon and an ALT-aware alignment approach, but at this time, we cannot get Pilon to properly polish the alternate contigs. We acknowledge that haplotype-aware assemblies are becoming the standard; however, we have a FALCON assembly completed with current standards, and we have supported many of our findings over several primate species.

Reply:

I am not sure if I agree here. Wouldn't it be possible to only take the primary contigs of the assembly and then treat it as 'pseudo-haploid'? That should alleviate the problem of multiple mappings of short reads and also enable a polishing step.

I am aware that genome assembly takes a lot of time and resources, thus I don't think it will be necessary to be repeated here. I'd just be interested if the results would change at all.

5. Page 5, last paragraph: '... that represents 2.95 Gb of the euchromatin ...' How was determined whether these 2.95 Gb of sequence belonged only to euchromatin? Please clarify.

Authors' response: To be accurate, we replaced "euchromatin" with "chromosomes".

6. Page 5, first paragraph, last sentence: The statement 'anchor scaffolds to chromosomes' might be more accurate by replacing 'chromosomes' with 'chromosome models'.

Authors' response: As suggested by the reviewer, we have replaced "chromosome" with "chromosome models".

7. Figure 2B: The spikes in closed gap lengths are intriguing. Is there an explanation for these?

Authors' response: In primate genomes, the distribution of gaps in assemblies is not random (Figure R2) and their sizes are often represented as approximations because they are not accurately known.

In GRCh38, for example, there are often gaps of 10,000 bp (n = 48), 50,000 bp (n = 169) and 100,000 bp (n = 20). Gaps of these three sizes alone account for 42% of all gaps in GRCh38 (237 of 559). The same pattern can be observed in rheMac8, except for small gaps (<100 bp in size), and there are often gaps in 100 bp, 100-1000 bp and 50,000 bp. These modal features are likely due to partially closed gaps.

Figure R2. The distribution of gap size in GRCh38 and rheMac8.

8. Page 8, first paragraph '... we also mapped 50-fold Illumina short read data ...': Is this the same data which was used in the Pilon polishing step? If so, isn't it surprising that there are any errors left?

Authors' response: Even with the best assemblies and polishing strategies, long-read assemblies still contain indel errors (Watson et al. 2019; Koren et al. 2019). The newest Sequel II HiFi protocol, which produces accurate long reads of QV 20+, cannot remove all of these assembly errors (Vollger et al. 2019). Although the error model of PacBio sequencing is random, homopolymer indels account for most of the remaining errors because of uncertainty introduced by random errors overlapped in homopolymer runs (e.g., "AAAAA..."). Pilon is also not effective in regions where short reads do not map uniquely, such as segmental duplications, regardless of sequencing depth or the number of polishing rounds. Although improvements in sequencing technology will ultimately remove many of these errors, this assembly represents current best practices.

9. Page 9, identification of SVs: There is no information on how SVs have been called (tools for SNV calling are mentioned). I suppose these were called via read mapping and Sniffles? Please clarify.

Authors' response: Variants were called from aligned contigs using a method developed for this purpose (Chaisson, et al. 2015). Briefly, the alignment CIGAR string for each contig is traversed, nearby small insertions and deletions are merged, and the resulting reference or contig gap is output as an insertion or a deletion (respectively).

Reply:

This is unclear to me. On line 195, you cite Sedlazeck et al. 2018, which does describe a read mapper and SV detection via read mapping. You don't describe any tool based on Chaisson et al. 2015 in the methods. Please clarify.

10. Page 10, last paragraph: Again, there is no information on how analyses (in this case the validation of SVs) were done.

Authors' response: Genotyping of the 53,916 SVs in rheMacS was conducted with Illumina deep sequencing data (HiSeq X10; PE read, >50-fold depth) in five unrelated Chinese rhesus monkeys using SVTyper (Chiang, et al. 2015). By inputting breakpoints of SVs, the WGS data was mapped to rheMac8, and then SVTyper infers genotypes at each site based on a Bayesian algorithm. Because SVTyper cannot genotype insertions directly, we obtained insertion calls by mapping the WGS reads to rheMacS, and the called deletions were regarded as insertions. According to the genotyping results, we counted how many SVs could be genotyped (validated) by the NGS data. We provided brief descriptions in the revised text, and a full description is included in Methods.

Reply:

In this sentence: "Because SVTyper cannot genotype insertions directly, we obtained insertion..." it is unclear which WGS reads you mean. If I understood correctly (based on your answer to 9.), you identified 53,916 SVs by aligning the rheMacS assembly to the rheMac8 assembly. I am not sure then why aligning data from unrelated individuals is used to validate insertions in the rheMacS individual,

since these could be unique to that individual. Please clarify.

11. Figure 3F: The depiction of variant validation with Sanger is very misleading / wrong. The variant is reportedly 4.3 kb of size, while the deletion in the sanger sequence is only a little over 100 bp. Please clarify.

Authors' response: In Figure 3F, we only showed the sequences (100 bp flanking sequences) surrounding the breakpoints of the deletion, and this was done by Sanger sequencing. However, as suggested by Reviewer #3, this case is not a good one to show the advantage of calling SVs using long-read sequencing because with proper calling tools, the short-read NGS data may also be able to call this SV. We therefore removed this case in the revised manuscript.

12. Page 12, last paragraph: The sentence: 'Although no functional category reached statistical significance, we did observe several interesting functional categories among the top-ranked categories.' essentially contains no information. It is not clear whether 'top-ranked categories' are meaningful in a scientific sense. Please adjust accordingly. I also suggest replacing the term 'hypothesize' with 'speculate'.

Authors' response: In the revised manuscript, we updated the phrase to "top 5% (10/208) categories". We also used the term "speculate" to replace "hypothesize" as suggested by reviewer.

13. Figure 4B: The synteny labelling is not clear to me. For example, if the color labels correspond to ape chromosomes, why is the second (dark green) macaque chromosome not 3 then?

Authors' response: The chromosome numbers are different among different primate species. Chromosome-2 of macaque is homologous to Chromosome-3 of apes and human (dark green) (Rogers, et al. 2006). As suggested by Reviewers #1 and #3, we moved it to supplementary materials.

Reply:

If the macaque chromosome 2 is homologous to chromosome 3 of apes, why is it not also chromosome 3 then? Or is it chromosome 2 because it is the second largest chromosome? My apologies if this is a very basic question.

14. Page 15, 'ASSVS in gene-coding regions', 3rd line '... and the other 7 were false (not ape-specific)': Does this mean that the variant was correctly called but was just not ape-specific? Or was it erroneously called in the first place?

Authors' response: All SVs mentioned in the ASSV section are correctly called—the "true" or "false" refers to whether they are ape specific. To be clear, we have added more explanation in the revised manuscript.

15. Page 15, 'ASSVs in gene-coding regions': Please explain 'nonsense-mediated decay' and provide references.

Authors' response: Nonsense-mediated decay (NMD) is a surveillance pathway that exists in all eukaryotes. Its main function is to reduce errors in gene expression by eliminating mRNA transcripts that contain premature stop codons (Baker and Parker 2004). Translation of these aberrant mRNAs could, in some cases, lead to deleterious gain-of-function or dominant-negative activity of the resulting proteins (Chang, et al. 2007). Given some NMD transcripts can be translated to functional proteins, VEP annotates those SVs located in the NMD transcripts as NMD transcript variants and coding sequence variants. We added references in the revised manuscript.

Reply:

Thank you for the clarification.

16. Page 20, last paragraph, sentence 'In contrast, SVs in the noncoding regions (together with other genetic changes such as SNVs) are under much less constraint, and they can provide more resources for phenotypic evolution.': This sentence is not clear to me; how can SVs in noncoding regions both be under less constraint (i.e.; leading to fewer phenotypic changes) and provide more resources for phenotypic evolution (i.e.; leading to more phenotypic changes). Please clarify and also add references to support your statement.

Authors' response: Variants in the noncoding regions are under much less constraint than coding regions. Less constraint means it is easier to accumulate mutations than coding regions under natural selection, and therefore would provide more resources for phenotypic evolution because they are unlikely to be immediately eliminated from the population (Bird, et al. 2007). However, as Reviewer #1 suggested, this discussion is not new and not very helpful for presenting our results, so we removed these sentences in the revised manuscript and rewrote this part.

17. Page 20, following the sentence above: Are 80 % of all SVs deletions? Or just the ape-specific ones? If the former, then this could also constitute a technical bias (deletions might be easier to discover) rather than a biological reason. Also, this seems a bit counterintuitive given that transposable element insertions are an important source of SVs. Please discuss these possibilities.

Authors' response: The 80% ASSVs are deletions (B in revised text) not all SVs. This was changed accordingly in the revised manuscript.

18. Page 21, first paragraph: Why were only insertions and deletions considered? Please clarify.

Authors' response: There were two reasons for considering only insertions and deletions. First, they account for more than 98% SVs in the assemblies, and smartie-sv is good for calling these two types. Second, because we used the NGS-based genome data of gibbon and marmoset to identify SVs occurred in the ape lineage, i.e., the ASSVs, it would introduce ambiguity in calling the other SV types such as duplications and translocations. Indeed, these SV types may also contribute to phenotypic evolution in primates, and they are worth studying in the future.

19. Page 22, second paragraph, PCR and Sanger sequencing of 25 coding ASSVs: To infer from these 25 SVs that the majority of the identified novel SVs are true is likely an overestimation, especially since SVs in coding vs. non-coding regions usually exhibit different features (e.g. SVs in tandem repeat arrays are more common in coding regions, but also are more likely false-positives). This should be taken into consideration.

Authors' response: We agree with this point. In the revised manuscript, we rephrased the sentence to avoid overestimation.

20. Page 25, Gap closure: How many Ns in a run were treated as a gap?

Authors' response: A region consisting of continuous runs of Ns ($N > 1$) in the rheMac8 chromosomes was defined as a gap. We have added this information in the revised text.

Reviewer #3 (Remarks to the Author):

Summary: In this revision, all of my major concerns were addressed by the authors. I still have several concerns with the discussion of the data that require revision, but I have more confidence in the results of the ASSV analysis.

On Tandem Repeats: The authors identified 30 candidate tandem repeat expansions of ASSVs but have not reported these in the manuscript. I understand that the determination of the exact mechanisms of SV formation require dedicated effort; however, I am satisfied if the authors merely report that 30 ASSVs appear to consist of tandem repeats but further efforts are needed to validate them.

Authors' response: We reported 30 tandem duplications, not tandem repeat expansions, which is worthy of further investigation in the future. As suggested by the reviewer, we added this information in the revised version (see Supplementary Table 20)

Line 416: The authors state in their response that they also used Sanger sequencing to validate 20 out of 22 non-coding ASSVs (with 2 being difficult to confirm due to their completely repetitive nature). Despite this being a good validation of their dataset, I do not see reference to this in the text. Such descriptions do not need to be lengthy, but it would be important to make clearer reference to this in the text, particularly since non-coding regions are predicted to consist of the majority of SVs in multi-cellular Eukaryotes.

Authors' response: We added the descriptions of validation of noncoding ASSVs in the discussion section of the revised manuscript.

Line 456: As written, this sentence is confusing to the reader. I would prefer to see an addition similar to the authors' response to my point regarding the comparison of short-read to long-read assemblies here. I would be satisfied if the authors note this point as a potential discrepancy in their comparisons and that a long-read assembly for marmoset is needed for more accurate comparisons.

Authors' response: As suggested by the reviewer, we added the description of the quality discrepancy among the two reference genomes and the need for long-read genome assemblies of marmoset and gibbon.

Reviewer #4 (who co-reviewed the first version of the manuscript with Reviewer 2)
(Remarks to the Author):

Detailed review:

1. Page 3, end of first paragraph: 'With so many gaps, it has been difficult to

systematically identify structural variants (SVs) in the macaque genome,..’ The relationship between gaps and reliable SV calling might not be clear to the reader. Please support this statement with a reference and maybe a brief explanation.

Authors’ response: Following the reviewer’s comment, we rephrased the sentence. Basically, the poor assembly quality of the published draft genome of Chinese rhesus macaque (Yan, et al. 2011) made it difficult to systematically identify SVs because of its poor contiguity (fragmentation) and incompleteness (many gaps). The misassemblies and missing regions of an NGS-based assembly most often stem from its limitation for assembly of repetitive DNA, including retrotransposons and segmental duplications (Alkan, et al. 2011). We have added a brief explanation and the relevant reference (Alkan, et al. 2011) in the revised text.

Reply:

The relationship between poor contiguity (many gaps) and SV calling is still not explained. To my knowledge, the problem lies in the fact that if an SV breakpoint lies in an assembly gap, it cannot be detected (correctly). Additionally, assembly gaps are often caused by repetitive DNA, which in turn promotes the formation of SV. Maybe you can change the sentence accordingly.

Authors’ response: Yes, many large tandem repeats and segmental duplications are still too large to assemble, and they also promote SV formation or are SVs themselves. With detection methods based on assemblies, we cannot detect SVs in regions where the assembly fails. This leaves many large SVs undetected. Improvements to long-read sequencing, which are actively being developed at PacBio and ONT, will help to resolve these regions and recover the SVs within them. We have rephrased the sentence in the revised version.

4. Page 4, results, line 4: Is there a particular reason for why ‘FALCON’ instead of ‘FALCON UNZIP’ was used? The latter results in a diploid assembly and would thus potentially be more accurate. Given that this is a diploid organism, heterozygous SVs could result in chimeric / erroneous contigs, impeding their detection.

Authors’ response: Mapping short reads to a FALCON UNZIP assembly fails because there are two representations of many loci, and short-read mappers do not handle alternate loci well. For example, this complicates polishing procedures using Pilon, which is absolutely essential for quality. With another sample, we have been experimenting with polishing UNZIP assemblies using Pilon and an ALT-aware alignment approach, but at this time, we cannot get Pilon to properly polish the alternate contigs. We acknowledge that haplotype-aware assemblies are becoming the standard; however, we have a FALCON assembly completed with current standards, and we have supported many of our findings over several primate species.

Reply:

I am not sure if I agree here. Wouldn’t it be possible to only take the primary contigs

of the assembly and then treat it as ‘pseudo-haploid’? That should alleviate the problem of multiple mappings of short reads and also enable a polishing step.

I am aware that genome assembly takes a lot of time and resources, thus I don’t think it will be necessary to be repeated here. I’d just be interested if the results would change at all.

Authors’ response: FALCON UNZIP gives a set of primary contigs (p-contigs) and alternate haplotigs (h-contigs), and yes, it would be possible to polish just the p-contigs and use the h-contigs as a set of lower-quality alternates. However, the p-contigs are not the same as a FALCON assembly. Without a guaranteed polishing method, it would have been a big risk to take to build the first Chinese macaque reference knowing there would be significant quality challenges that may or may not have a good workaround.

9. Page 9, identification of SVs: There is no information on how SVs have been called (tools for SNV calling are mentioned). I suppose these were called via read mapping and Sniffles? Please clarify.

Authors’ response: Variants were called from aligned contigs using a method developed for this purpose (Chaisson, et al. 2015). Briefly, the alignment CIGAR string for each contig is traversed, nearby small insertions and deletions are merged, and the resulting reference or contig gap is output as an insertion or a deletion (respectively).

Reply:

This is unclear to me. On line 195, you cite Sedlazeck et al. 2018, which does describe a read mapper and SV detection via read mapping. You don’t describe any tool based on Chaisson et al. 2015 in the methods. Please clarify.

Authors’ response: For identification of SVs in the rheMacS assembly, we used a read-mapping approach NGLMR+Sniffles (Sedlazeck, et al. 2018) by mapping the PacBio long reads to the current reference genome rheMac8. For identification of ASSVs, we used smartie-sv, a tool developed by Chaisson et al. 2015 and modified by Kronenberg et al. 2018, through pairwise comparison between rheMacS and the ape genomes. We added the Chaisson et al. 2015 citation in the smartie-sv part. To be clear, we rephrased the description of rheMacS SVs identification in the revised text.

10. Page 10, last paragraph: Again, there is no information on how analyses (in this case the validation of SVs) were done.

Authors’ response: Genotyping of the 53,916 SVs in rheMacS was conducted with Illumina deep sequencing data (Hiseq X10; PE read, >50-fold depth) in five unrelated Chinese rhesus monkeys using SVTyper (Chiang, et al. 2015). By inputting breakpoints of SVs, the WGS data was mapped to rheMac8, and then SVTyper infers genotypes at each site based on a Bayesian algorithm. Because SVTyper cannot genotype insertions directly, we obtained insertion calls by mapping the WGS reads to

rheMacS, and the called deletions were regarded as insertions. According to the genotyping results, we counted how many SVs could be genotyped (validated) by the NGS data. We provided brief descriptions in the revised text, and a full description is included in Methods.

Reply:

In this sentence: “Because SVTyper cannot genotype insertions directly, we obtained insertion...” it is unclear which WGS reads you mean. If I understood correctly (based on your answer to 9.), you identified 53,916 SVs by aligning the rheMacS assembly to the rheMac8 assembly. I am not sure then why aligning data from unrelated individuals is used to validate insertions in the rheMacS individual, since these could be unique to that individual. Please clarify.

*Authors’ response: The unrelated individuals were used for genotyping the identified SVs in rheMacS by mapping the **ILLUMINA short reads** of five unrelated macaques to rheMac8 (deletion calling) and rheMacS (insertion calling) using SVTyper. The presence of an SV in the five unrelated macaques gives more confidence for it being true, and at the same time, can give a rough estimation of SV frequency. We rephrased the sentence in the revised text.*

13. Figure 4B: The synteny labelling is not clear to me. For example, if the color labels correspond to ape chromosomes, why is the second (dark green) macaque chromosome not 3 then?

Authors’ response: The chromosome numbers are different among different primate species. Chromosome-2 of macaque is homologous to Chromosome-3 of apes and human (dark green) (Rogers, et al. 2006). As suggested by Reviewers #1 and #3, we moved it to supplementary materials.

Reply:

If the macaque chromosome 2 is homologous to chromosome 3 of apes, why is it not also chromosome 3 then? Or is it chromosome 2 because it is the second largest chromosome? My apologies if this is a very basic question.

Authors’ response: This is the most commonly used rule of chromosome nomenclature in primates so that among-species comparison would be easier (Rogers et al. 2006).

Reference

Chaisson, M. J., et al. Resolving the complexity of the human genome using single-molecule sequencing. *Nature*. 517(7536):608-11 (2015).
Rogers, J., et al. An initial genetic linkage map of the rhesus macaque (*Macaca mulatta*) genome using human microsatellite loci. *Genomics*. 87(1):30-38 (2006).
Sedlazeck, F. J., et al. Accurate detection of complex structural variations using single-molecule

sequencing. *Nature Methods*. 15(6):461-+ (2018).

Kronenberg, Zev N., et al. High-resolution comparative analysis of great ape genomes. *Science*. 360(6393):1085-+ (2018).

REVIEWERS' COMMENTS:

Reviewer #4 (Remarks to the Author):

The authors have now addressed all my comments and clarified all issues. I would hence recommend the manuscript for publication.